# Local activation of focal adhesion kinase orchestrates the positioning of presynaptic scaffold proteins and Ca²⁺ signalling to control glucose-dependent insulin secretion

**Dillon Jevon[1†], Kylie Deng[1†], Nicole Hallahan[1†], Krish Kumar[1], Jason Tong[1], Wan Jun Gan[1], Clara Tran[2], Marcela Bilek[2,3,4], Peter Thorn[1]***

[1]Charles Perkins Centre, School of Medical Sciences, University of Sydney, Sydney, Australia; [2]School of Physics, University of Sydney, Sydney, Australia; [3]School of Aerospace, Mechanical and Mechatronic Engineering, University of Sydney, Sydney, Australia; [4]Sydney Nanoscience Institute, University of Sydney, Sydney, Australia

**\*For correspondence:**
p.thorn@sydney.edu.au

[†]These authors contributed equally to this work

**Competing interest:** The authors declare that no competing interests exist.

**Abstract** A developing understanding suggests that spatial compartmentalisation in pancreatic β cells is critical in controlling insulin secretion. To investigate the mechanisms, we have developed live-cell subcellular imaging methods using the mouse organotypic pancreatic slice. We demonstrate that the organotypic pancreatic slice, when compared with isolated islets, preserves intact β-cell structure, and enhances glucose-dependent Ca²⁺ responses and insulin secretion. Using the slice technique, we have discovered the essential role of local activation of integrins and the downstream component, focal adhesion kinase (FAK), in regulating β cells. Integrins and FAK are exclusively activated at the β-cell capillary interface and using in situ and in vitro models we show their activation both positions presynaptic scaffold proteins, like ELKS and liprin, and regulates glucose-dependent Ca²⁺ responses and insulin secretion. We conclude that FAK orchestrates the final steps of glucose-dependent insulin secretion within the restricted domain where β-cell contact the islet capillaries.

## Editor's evaluation

The authors study insulin secretion in acutely prepared pancreatic slices and find that it is remarkably different from what is observed in isolated islets. In particular, the work shows that polar differentiation and higher stimulus-secretion coupling is caused by integrin-mediated and local establishment of effective release sites where the beta cells contact the capillaries, involving concentration of active zone proteins and clustering of calcium channels. The findings are important and should be of significant interest to the part of readership with an interest in the regulation of exocytosis in general and insulin secretion in particular.

## Introduction

The intrinsic stimulus secretion coupling cascade in pancreatic β cells is well understood through extensive in vitro experimentation (*Rorsman and Ashcroft, 2018*). However, within the native islets of Langerhans numerous external factors intersect with this signal cascade to further control secretion (*Lammert and Thorn, 2020*; *Meda, 2013*). The impact of some factors, such as gap junctions between endocrine cells, is well understood (*Benninger et al., 2011*). Less well understood is the

impact of the islet microenvironment on β-cell structural organisation and function (*Lammert and Thorn, 2020*) and how this intersects with the known stimulus secretion pathways.

Accumulating evidence suggests that the region where β-cell contact the islet capillaries is specialised for secretion (*Gan et al., 2017*; *Low et al., 2014*). β cells, within intact islets, make a discrete point of contact with the extracellular matrix (ECM) that surrounds the capillaries. This point of contact is the target for insulin granule fusion (*Low et al., 2014*) and is enriched in presynaptic scaffold proteins, like liprin and ELKS and therefore has characteristics analogous to a neuronal presynaptic domain (*Deng and Thorn, 2022*; *Lammert and Thorn, 2020*; *Low et al., 2014*; *Ohara-Imaizumi et al., 2019*; *Ohara-Imaizumi et al., 2005*). Recapitulating this domain by culture of β cells on ECM-coated dishes shows that local activation of integrins is the target for insulin granule fusion (*Gan et al., 2018*) and local control of microtubules regulates these secretory hot spots (*Trogden et al., 2021*). Although the mechanisms are not known this work suggests that presynaptic scaffold proteins, and perhaps microtubules, control granule targeting to this capillary interface.

Just like neurotransmitter release, $Ca^{2+}$ is the dominant regulator of insulin secretion principally by $Ca^{2+}$ entry through voltage-sensitive $Ca^{2+}$ channels (*Schulla et al., 2003*). We know from other systems that the location of $Ca^{2+}$ channels relative to sites of granule fusion is critical to stimulus secretion coupling (*Nanou and Catterall, 2018*; *Stanley, 1997*). $Ca^{2+}$ channels are typically regulated by intracellular $Ca^{2+}$ concentrations leading to positive and negative feedback to control channel opening (*Zühlke et al., 1999*). These actions control the amplitude and temporal kinetics of local subcellular $Ca^{2+}$ concentrations which in turn regulate exocytosis (*Nanou and Catterall, 2018*). In neurones, a presynaptic scaffold protein complex tethers synaptic vesicles and collocates $Ca^{2+}$ channels to the presynaptic domain (*Südhof, 2012*); whether similar mechanisms exist at the capillary interface of β cells is unknown.

In β cells there is functional, in vitro, evidence for close association of Cav1.2, $Ca^{2+}$ channels and insulin granule exocytosis (*Bokvist et al., 1995*; *Gandasi et al., 2017*; *Pertusa et al., 1999*) and structural evidence for protein links between the Cav1.2 channels and syntaxin 1A (*Wiser et al., 1999*); a SNARE protein required for granule fusion. This evidence is based on single, isolated β cells where capillary contacts are not present and the normal environmental cues of the islets are lost. Immunostaining β cells in the more intact environment of pancreatic slices shows that syntaxin 1A (*Low et al., 2014*) has an even distribution across the β-cell plasma membrane and no enrichment at the capillary interface. This evidence therefore discounts a simple model where insulin secretion is regulated by colocalisation of syntaxin 1A and Cav1.2 at the capillary interface. Instead, there is recent evidence that the scaffold protein ELKS interacts with the β subunit of the $Ca^{2+}$ channel (*Ohara-Imaizumi et al., 2019*). Furthermore, although the work was carried out in isolated islets, which lack capillaries, there was evidence that the coupling between ELKS and the β subunit enhanced the $Ca^{2+}$ response at residual capillary structures (*Ohara-Imaizumi et al., 2019*), consistent with the idea that localised synaptic-like regulation of $Ca^{2+}$ and exocytosis might exist in β cells (*Deng and Thorn, 2022*). However, the mechanisms that organise and control the positioning of these presynaptic scaffold proteins is unknown.

The emerging picture therefore is that spatial compartmentalisation is a key attribute of stimulus–secretion coupling in pancreatic β cells. The capillary interface of β cells is a region enriched in presynaptic scaffold proteins, is the target for insulin granule fusion and might be a region where $Ca^{2+}$ channels are regulated. However, progress in this area is hampered by the difficulties in imaging single β cells within the islet environment.

To this end, the pancreatic slice is an important advance with a closer preservation to native islet structure than isolated islets (*Gan et al., 2017*; *Meneghel-Rozzo et al., 2004*). Analogous to organotypic brain slices, pancreatic slices maintain complex cell-to-cell arrangements that are likely to be important for overall islet control such as an intact islet capillary bed (*Cohrs et al., 2017*; *Gan et al., 2017*). In addition, the local microenvironment around each endocrine cell is maintained, with each cell contacting the capillary and other endocrine cells. This promotes a distinct subcellular polarisation in β cells that is likely to impact on cell function (*Gan et al., 2017*) with recent evidence the same organisation is present in rodent and human islets (*Cottle et al., 2021*). To date the pancreatic slice has been used in fixed-cell studies (e.g. *Gan et al., 2017*) and functional studies, for example looking at coordination of $Ca^{2+}$ responses in β cells across the islet (*Stožer et al., 2013*; *Stožer et al., 2021*). In principle, the slice is the ideal platform for live-cell subcellular studies of the effects of β-cell

organisation on glucose-dependent responses. However, preservation of function in slices has proved difficult and to date single-cell, live-cell work has relied on isolated islets (eg: *Low et al., 2014*; *Ohara-Imaizumi et al., 2019*) where capillaries are damaged and fragmented (*Irving-Rodgers et al., 2014*; *Lukinius et al., 1995*).

Here, we have developed the pancreatic slice preparation for live-cell subcellular imaging of β-cell responses to glucose. Compared to isolated islets we show: slices demonstrate local activation of integrins and focal adhesions at the capillary interface of β cells, preserve enrichment of presynaptic scaffold proteins and have highly targeted insulin granule fusion to the capillary interface, fast $Ca^{2+}$ spikes at low glucose concentrations and fast $Ca^{2+}$ kinetics in response to glucose elevation with very fast intracellular $Ca^{2+}$ waves that originate at the capillary interface.

We test for a role of contact with the capillary ECM using a range of interventions to block integrins and focal adhesion kinase (FAK) all of which consistently inhibit glucose-dependent $Ca^{2+}$ responses and insulin secretion and disrupt the positioning of the presynaptic scaffold proteins ELKS and liprin. Importantly, we show high potassium-induced secretion and $Ca^{2+}$ responses are not affected by these interventions demonstrating the integrin/FAK pathway is a key and selective mediator of glucose control.

Together our data demonstrate that FAK is a master regulator of glucose-induced insulin secretion that controls the positioning of presynaptic scaffold proteins and shapes the $Ca^{2+}$ responses.

## Results

A striking characteristic of islets within slices is the preservation of the rich capillary bed (*Gan et al., 2017*) which contrasts with the loss of endothelial cells and capillaries in the more usual method of enzymatic islet isolation (*Lukinius et al., 1995*).

Using an immunostaining approach the distribution of ECM was markedly different between pancreatic slices and isolated islets (*Figure 1*). This is as expected, because endothelial cells are the only source of intra-islet ECM (*Nikolova et al., 2006*). In the isolated islets laminin was not associated with any specific structure but instead was dotted throughout the islets (*Figure 1A*). In pancreatic slices the ECM, identified by laminin, was strongly enriched around the islet capillaries and in the islet capsule (*Figure 1B*). Consistent with this disruption we observed a reduction in laminin area and a reduction in CD31 (endothelial cell marker) immunostaining (*Figure 1C*).

ECM activates integrin-mediated responses in β cells (*Gan et al., 2018*; *Parnaud et al., 2006*; *Rondas et al., 2012*), we therefore sought to define the subcellular responses in β cells in the two preparations. In slices, we observed tight alignment of integrin-β1 with laminin-stained capillaries (*Figure 1E*). Phosphorylated FAK (phospho-FAK) which provides a read out of focal adhesion activity (*Rondas et al., 2012*) was also enriched at the capillary interface (*Figure 1F*).

Experiments show that the capillary interface of pancreatic β cells has similarities to the presynaptic domain of neurons including the enrichment of synaptic scaffold proteins, like liprin and ELKS (*Gan et al., 2017*; *Lammert and Thorn, 2020*; *Low et al., 2014*; *Ohara-Imaizumi et al., 2005*). In vitro, we have previously shown that local integrin activation is a primary factor in causing the clustering of liprin (*Gan et al., 2018*). Using immunostaining in pancreatic slices, as expected, liprin showed enrichment at the capillary interface (stained with laminin) and very little staining in other regions around the β cell (*Figure 1G*). The fluorescence intensity of laminin and liprin staining was quantified in four regions of interest around the β-cell periphery (*Figure 1H, I*). In slices, the region adjacent to the capillary had significantly higher staining for both laminin and liprin (*Figure 1H, I*) as illustrated in a heatmap (*Figure 1J*).

In isolated islets, consistent with the relative loss of ECM proteins (*Figure 1A*) we observed a misdistribution of integrin-β1 (*Figure 1K*), although interestingly, integrin-β1 was still present but was now all around the cells. Phospho-FAK was enriched at the residual capillaries (*Figure 1L*) but, using an area analysis, we observe a significant and approximately fivefold decrease in area occupied by phospho-FAK in the islets compared to slices (*Figure 1D*). These data show that the disruption in ECM in the isolated islets does affect the function of β cells, in this case, reducing focal adhesion activity, as measured by phosphorylation. Also disrupted was the positioning of liprin. In isolated islets, liprin was dispersed and located all around the β-cell surface (*Figure 1M*). In our analysis, we arbitrarily assigned the strongest laminin staining to region 1 (because, unlike the slice, we cannot readily identify the location of the capillaries). This alignment resulted in a significantly greater proportion of

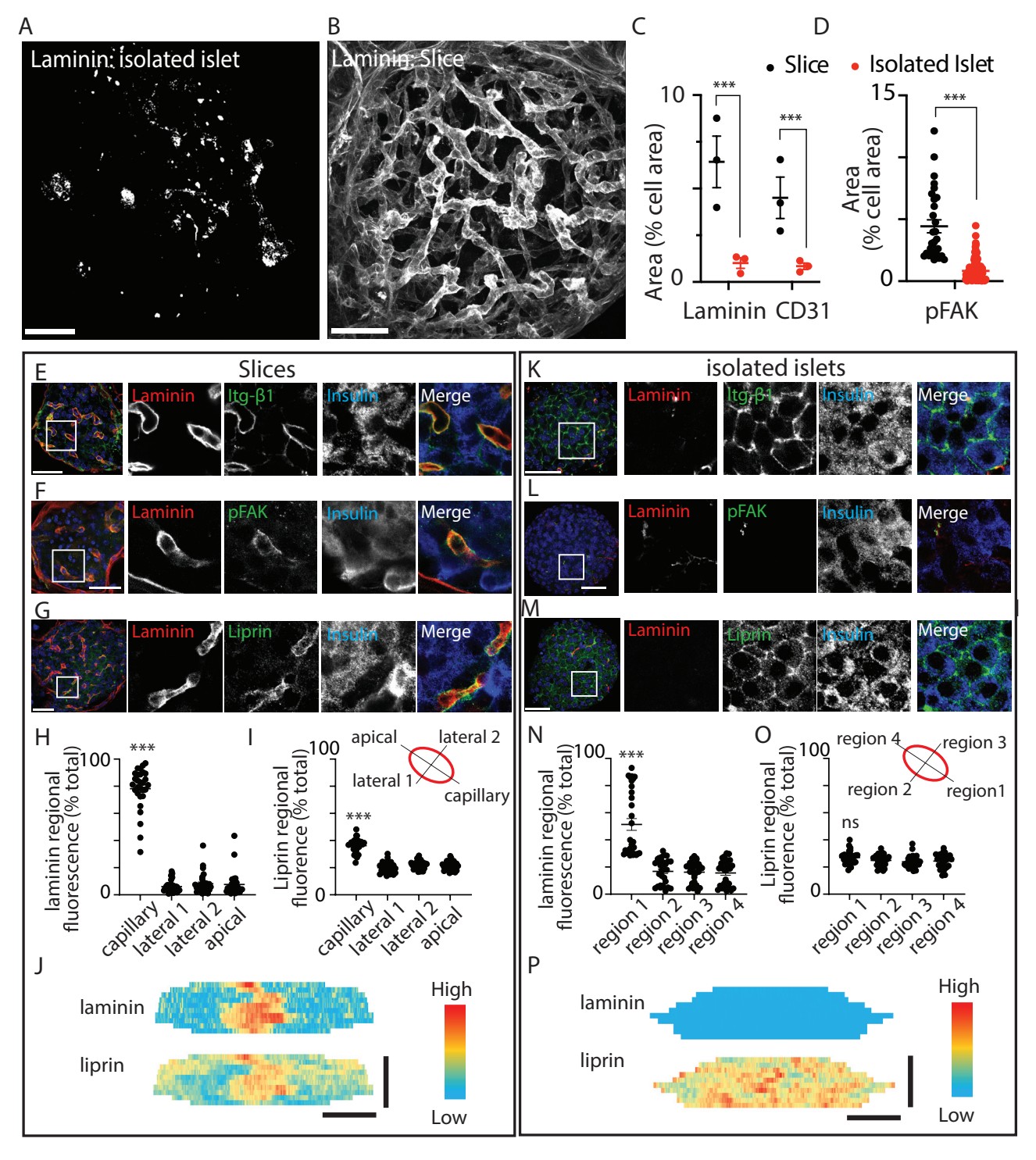

**Figure 1.** Pancreatic slices have an intact capillary bed. Integrin-β1, phosphorylated focal adhesion kinase (FAK), and liprin are enriched at the β-cell capillary interface. Pancreas slices and isolated islets were cultured overnight prior to fixation and immunostaining. Representative 3D projection of the extracellular matrix (ECM) protein laminin through (**A**) an isolated islet and (**B**) an islet within a slice. (**C**) Quantification of laminin and CD31 immunofluorescence, normalised to cell area (insulin + 4',6-diamidino-2-phenylindole (DAPI), see Materials and methods for details) in the corresponding Z-planes showed a significant loss of both proteins in isolated islets compared to slices ($n$ = 3 and 2–3 islets analysed per mouse, mean ± standard error of the mean (SEM), Student's $t$-test $p < 0.001$). Scale bar 40 µm. (**D**) Phospho-FAK immunostaining shows significantly reduced area (compared to total cell area, insulin + DAPI) in β cells in isolated islets ($n$ = 29 cells in slices 112 cells in islets, Student's $t$-test $p < 0.01$). Scale bar 40 µm. (**E–G**) Immunostaining in slices for integrin-β1, laminin, and liprin shows integrin and liprin are enriched at the capillary interface. Scale bar 40 µm. (**H, I**)

*Figure 1 continued on next page*

*Figure 1 continued*

Quantification of fluorescence in four regions of interest (~2 × 2 µm) places around individual β cells (see cartoon) showed significant liprin enrichment at the capillary interface region compared to the lateral and apical regions (one-way analysis of variance [ANOVA] showed significant differences across the regions ($F(3,112) = 2.421$; $p < 0.0001$, a Tukey post hoc comparison showed the capillary domain was brighter compared to each other domains ($p < 0.0001$), $n = 29$ cells, four slices, three mice). (**J**) Heatmaps of the total surface area of single β cells were prepared as previously described (*Gan et al., 2018*) by measuring fluorescence intensity (of liprin and laminin staining) along linescans drawn around the circumference of each cell, at each Z slice. Fluorescence intensity expressed on a pseudocolour scale (normalised to maximum fluorescence) shows the local enrichment of liprin coincident with laminin staining. Scale bar 5 µm. (**K–M**) In isolated islets, the capillaries are lost, as shown by lack of laminin staining, and liprin and integrin staining is now dispersed across the membrane. (**N, O**) In isolated islets, capillaries were absent and laminin staining weak, we therefore arbitrarily assigned region 1 as the region with maximal laminin staining. Even after this post-analysis alignment of the regions liprin staining was similar across all regions (one-way ANOVA no difference across the regions ($F(3,120) = 0.57$; $p = 0.054$, $n = 31$ cells, 4 islets, 3 mice). (**P**) Heatmaps with fluorescence intensity expressed on a pseudocolour scale shows weak laminin staining and relatively uniform liprin staining across the β-cell area. Scale bar 5 µm. * shows statistical significance at $p<0.05$; ** shows statistical significances at $p<0.01$; *** shows statistical significance at $p<0.001$.

The online version of this article includes the following source data and figure supplement(s) for figure 1:

**Source data 1.** pFAK area analysis.

**Source data 2.** CD31 area analysis.

**Source data 3.** Liprin distribution analysis.

**Figure supplement 1.** Immunostaining for E-cadherin and PAR3 in slices and isolated islets.

laminin staining in region 1 (*Figure 1N*) but the liprin staining was still evenly spread across all regions (*Figure 1O*) and illustrated in a heatmap (*Figure 1P*).

We also show that β-cell structure is affected, lateral β-cell contacts were still maintained, as shown by E-cadherin immunostaining but Par3, normally located in the apical region of β cells, away from the capillaries showed diffuse non-polar organisation in isolated islets (*Figure 1—figure supplement 1*).

We conclude that the organotypic slice preserves the secondary structure of the islet, such as the capillary bed and the polarised structure of the β cells. Functionally, this translates into a local positioning of integrins and the local activation of phospho-FAK, both of which are significantly disrupted in isolated islets. We therefore set out to use the slice as a platform to test the functional consequences of preserved β-cell structure and activation of the integrin/FAK pathway.

## Organotypic slices have significantly enhanced glucose-sensitive insulin secretion

In the experiments described from Figures 2–7 we compare responses of β cells in isolated islets with those from pancreatic slices. For both approaches we cultured the preparations overnight and recorded responses on the following day. In this way, the time in culture was exactly the same.

Our past work with isolated islets demonstrated that insulin granule fusion is targeted to the interface of the remnant capillaries in isolated islets (*Low et al., 2014*). The method uses a dye-tracing technique to identify in space and time the fusion of individual secretory granules induced by a step increase in glucose from 2.8 to 16.7 mM. Here, we repeat those findings and record each fusion event over a 20-min stimulus with high glucose (*Figure 2A*) but now, in parallel experiments, we compare the distribution of events obtained using pancreatic slices (*Figure 2B*). In both preparations targeting to the capillary interface of β cells is observed but in the slice preparation the targeting is significantly enhanced to the extent that nearly 80% of all granule fusion events occur in this region (*Figure 2C, D*). The greater precision in targeting of insulin granule fusion is consistent with the tight focus of phospho-FAK enrichment in slices (*Figure 1F*) and with previous in vitro reports that integrin activation drives granule targeting of granule fusion (*Gan et al., 2018*).

The granule fusion assay gives a quantitative measure of exocytosis but the low sample number of cells makes quantification of insulin secretion difficult and therefore led us to directly measure insulin secretion using a bulk secretion assay. Here, we demonstrate that pancreatic slices, compared to isolated islets have, significantly increased insulin secretion at all concentrations of glucose (*Figure 3A*). Furthermore, we observed insulin secretion at low glucose (2.8–5 mM) concentrations in slices that was not seen in isolated islets, and the overall $EC_{50}$ for glucose dose dependence was different (*Figure 3A*, $EC_{50}$ 10.2 mM for islets and 8.6 mM for slices). In control experiments, embedding isolated islets in agarose (the substrate used to embed slices) had no effect in insulin secretion (*Figure 3—figure supplement 1*). Furthermore, there was no difference in measured proinsulin

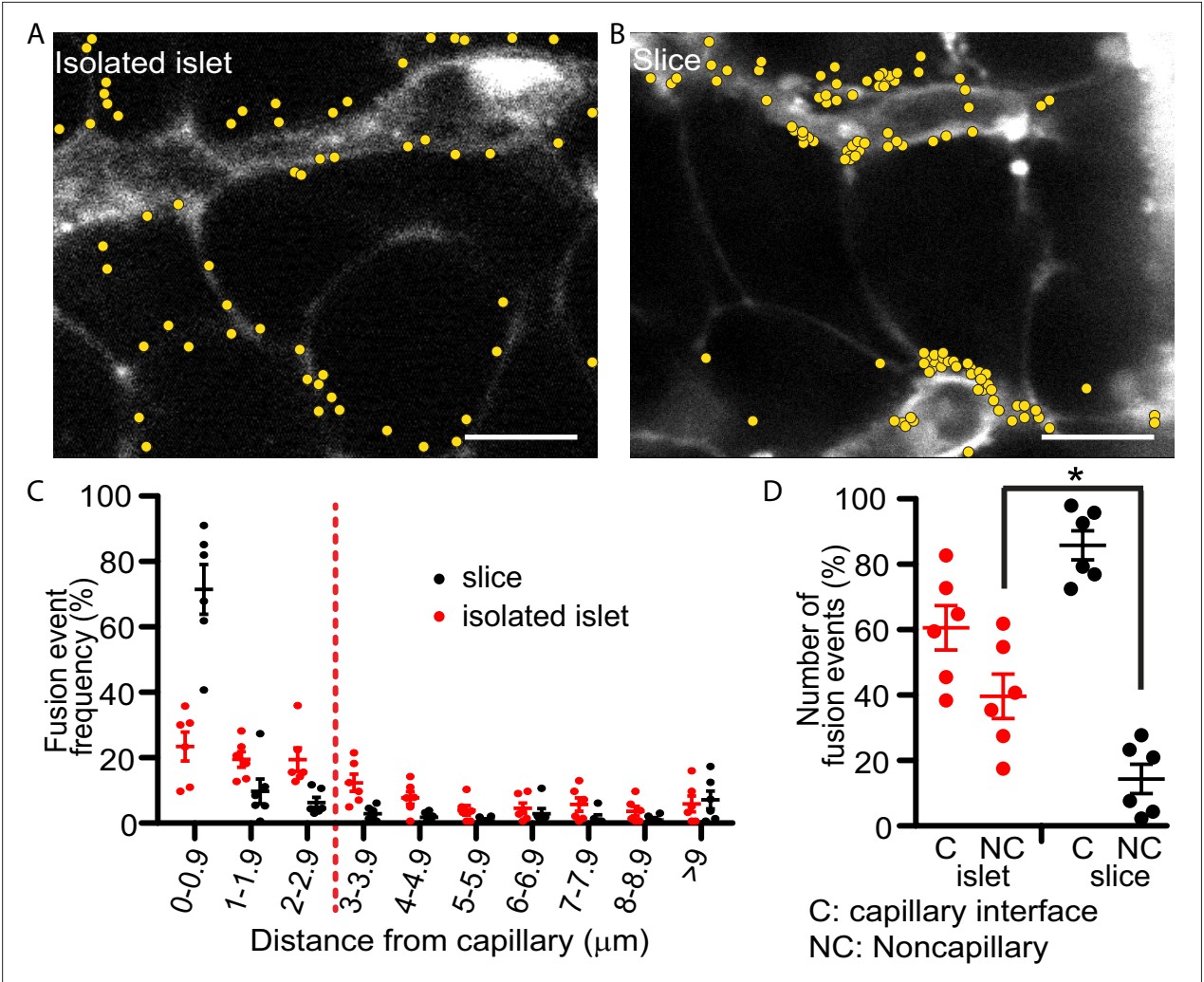

**Figure 2.** Glucose-stimulated insulin granule fusion in isolated islets and pancreas slices. (**A**) Isolated islets and (**B**) pancreas slices, bathed in an extracellular dye (sulforhodamine B, SRB). This dye outlines each cell and is enriched in the islet capillaries (**Low et al., 2013**) which are observed as large, elongated fluorescence structures and are fragmented in isolated islets but extended and continuous in slices. When cells were stimulated with 16.7 mM glucose to induce insulin granule fusion, which is recorded as the sudden and transient appearance of bright spots of fluorescence (**Low et al., 2014**). Continuous recording of two-photon images over 20 min of glucose stimulation led to many exocytic events, which were identified and marked on the images with yellow dots. (**C**) Slices ($n = 6$ slices) had a strong bias of fusion events towards the vasculature, while fusion events in isolated islets ($n = 6$ islets) were more spread out. (**D**) Fusion events in isolated islets and slices were classified as either occurring at the capillary face (<2.9 μm, **C**) or elsewhere on the cell membrane (>2.9 μm, NC). All data are represented as the mean ± standard error of the mean (SEM) ($n = 3$), significance determined by Student's $t$-tests, $p < 0.05$. * shows statistical significance at $p<0.05$.

secretion in slices compared to isolated islets, indicating that insulin processing was unchanged (**Figure 3—figure supplement 1**). Our values for insulin secretion in isolated islets are comparable with other reports (**Rorsman and Ashcroft, 2018**) and overall our data show that slices are more sensitive to glucose and secrete more insulin than isolated islets.

A more detailed interrogation of glucose-dependent control of insulin secretion segregates glucose action into two distinct routes: a trigger and an amplification pathway (**Gembal et al., 1992**; **Henquin, 2009**). The glucose triggering pathway includes the steps from glucose uptake, closure of $K_{ATP}$ channels, and the activation of voltage-dependent $Ca^{2+}$ channels and the subsequent exocytosis of insulin granules (**Rorsman and Ashcroft, 2018**). Less is known about the amplification pathway which is characterised by a glucose-dependent augmentation of insulin secretion (**Gembal et al., 1992**; **Henquin, 2009**) potentially by controlling granule transport and docking prior to fusion (**Ferdaoussi et al., 2015**). One approach to distinguish between the trigger and the amplification pathways

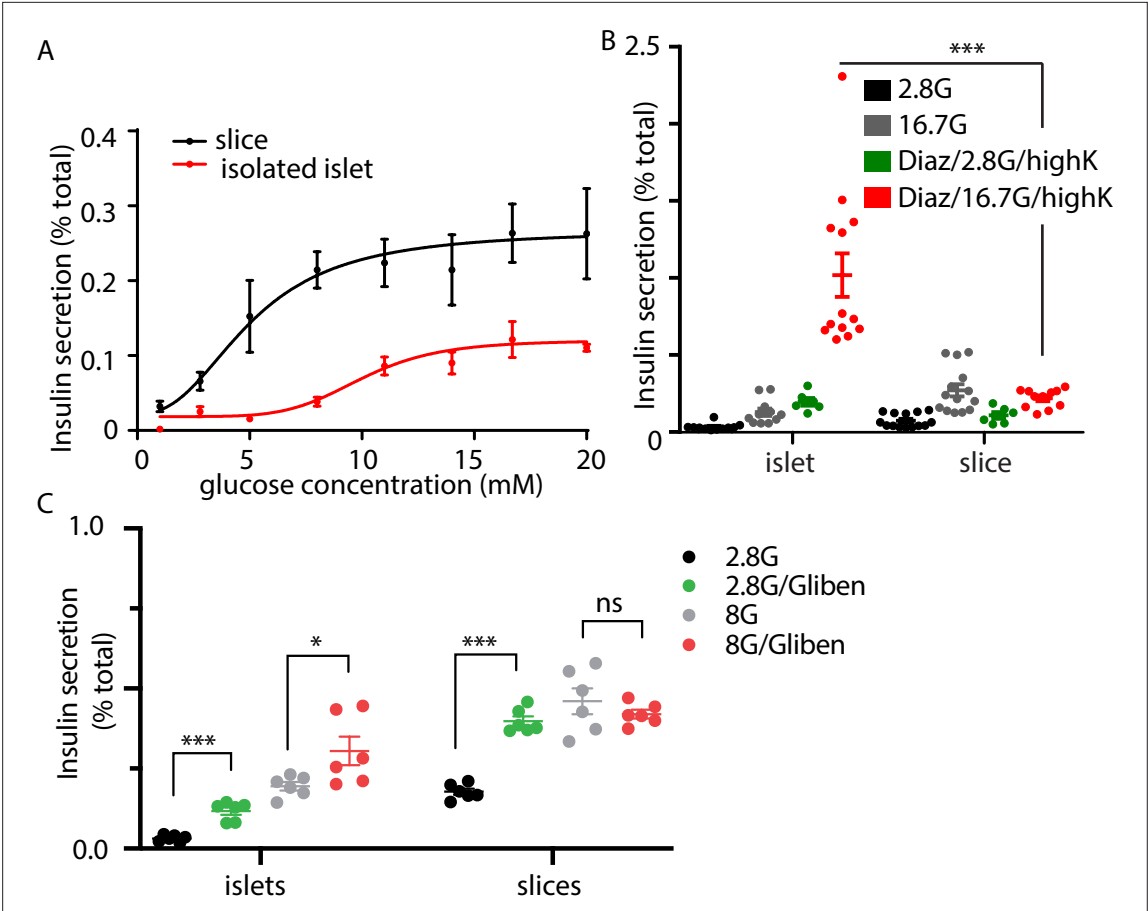

**Figure 3.** Glucose-stimulated secretion in isolated islets and pancreas slices. (**A**) Dose-dependent glucose-stimulated insulin secretion normalised to total cellular insulin content shows that isolated islets are less sensitive to low glucose concentration and secrete a lower proportion of their total content compared to islets in pancreas slices (*n* = 3–14 mice at each point). Two-way analysis of variance (ANOVA) showed significant main effects of islet preparation ($F_{(1, 76)}$ = 35.53; p < 0.0001) and glucose concentration ($F_{(7, 76)}$ = 8.657; p < 0.0001), but no significant interaction between the two factors ($F_{(7, 76)}$ = 1.314; p = 0.2553). The lines are non-linear best fit dose–response curves with a fitted $EC_{50}$ of 10.2 mM for islets and 8.6 mM for slices. (**B**) Islets and slices were incubated either with glucose alone at 2.8 or 16.7 mM glucose or in the presence of 250 μM diazoxide where secretion was subsequently stimulated by raising extracellular potassium. The response in the presence of diazoxide and 16.7 mM glucose, which reflects glucose amplification was significantly greater in islets compared to slices (*n* = 6–13 islets or slices from *n* = 3 animals, Student's *t*-test p < 0.001). (**C**) Pretreatment with the $K_{ATP}$ channel blocker, glibenclamide (2 μM) significantly increased insulin secretion measured at 2.8 mM glucose in both isolated islets and slices (Student's *t*-test p < 0.01, *n* = 3 mice for both conditions). In contrast, pretreatment with glibenclamide only increased insulin secretion measured at 8 mM in islets but not in slices (Student's *t*-test p < 0.05 for islets, p = 0.36 for slices, *n* = 3 mice for both conditions). * shows statistical significance at p<0.05; *** shows statistical significance at p<0.001.

The online version of this article includes the following source data and figure supplement(s) for figure 3:

**Figure supplement 1.** Insulin secretion in pancreatic slices and isolated islets.

**Figure supplement 1—source data 1.** Insulin secretion data.

uses diazoxide, a $K_{ATP}$ channel opener, to clamp the β-cell membrane potential negative prior to addition of glucose at different concentrations (**Gembal et al., 1992**). Glucose addition then does not cause insulin secretion, because of the presence of diazoxide, but secretion can be triggered by exposure to high potassium. Comparison of the responses at different glucose concentrations then defines glucose-dependent amplification (**Henquin, 2009**).

In our experiments, because glucose-dependent secretion was greater in pancreatic slices at all glucose concentrations (**Figure 3A**), we were anticipating that amplification would be larger. Surprisingly, our results showed the opposite and in fact glucose-dependent amplification was significantly larger in isolated islets compared with pancreatic slices (**Figure 3B**). This enhanced amplification

suggests the overall decrease in glucose-dependent secretion in isolated islets compared to slices (*Figure 3A*), must be due to reduced glucose-dependent triggering.

In additional experiments, we pretreated with the $K_{ATP}$ channel blocker glibenclamide (*Figure 3C*) which enhanced insulin secretion measured at 2.8 mM glucose in both isolated islets and in slices. Interestingly, at 8 mM glucose the addition of glibenclamide increased insulin secretion only in isolated islets (*Figure 3C*). This further supports the idea that glucose-dependent triggering is compromised in isolated islets and that $K_{ATP}$ block can overcome this deficit.

These results demonstrate that at least one component of the enhanced secretion seen in pancreatic slices is due to β-cell intrinsic differences. The mechanisms behind glucose-dependent amplification are not well understood and the increase in isolated islets are therefore difficult to study. However, the steps in glucose-dependent triggering are well understood and lead to $Ca^{2+}$ responses. Given the enhancement in secretion in pancreatic slices we set out to characterise this glucose triggered $Ca^{2+}$ signal in more detail.

## Glucose-dependent triggering: fast intracellular $Ca^{2+}$ waves characterise responses in pancreatic slices

The final step in the glucose-dependent triggering pathway is the entry of $Ca^{2+}$ through voltage-sensitive $Ca^{2+}$ channels that open at each action potential (*Rorsman and Ashcroft, 2018*). We chose to study intracellular $Ca^{2+}$ responses using the genetically encoded $Ca^{2+}$ probe, GCAMP6s which was expressed in the β cells using knock-in *Ins1Cre* mice (*Thorens et al., 2015*). The β cells were imaged using multiphoton microscopy and the responses across a range of glucose concentrations measured.

In slices (*Figure 4A–D*) and isolated islets (*Figure 4E–H*) we observed characteristic large responses to high glucose concentrations of 16.7 mM. When recording from different cells in the field of view we usually observed synchronous responses across many cells (*Figure 4D, H*) indicating that in both preparations the cells are functionally coupled through gap junctions (*Benninger et al., 2011*). In these recordings the $Ca^{2+}$ responses from β cells within slices typically showed pulsatile activity even at 2.8 mM glucose (*Figure 4D*), which is consistent with the observations of insulin secretion at this low glucose concentration (*Figure 3A*), and we also observed rapid pulsing of $Ca^{2+}$ at the beginning of the high glucose-induced responses in slices, consistent with enhanced excitability.

We next recorded $Ca^{2+}$ responses and determined the time when high glucose arrived at the cells by including a fluorescent probe in the high glucose solution (*Figure 5—figure supplement 1*). $Ca^{2+}$ responses in slices (*Figure 5A, B*) were apparently initiated almost simultaneously with the addition of 16.7 mM glucose indicating that these large responses are triggered by even small elevations in the concentration of glucose. In contrast, in isolated islets the $Ca^{2+}$ responses occurred with a consistent delay after the addition of glucose (*Figure 5C, D*). Comparison of the parameters of the global $Ca^{2+}$ responses to 16.7 mM glucose in slices with those in isolated islets (*Figure 5F, G, H1*) shows the time to peak mean was significantly shorter (*Figure 5I*) and the frequency distribution was shifted to shorter times in slices (*Figure 5J*).

As before (*Figure 4*) we consistently observed pulsatile $Ca^{2+}$ activity at 2.8 mM glucose (*Figure 5A, B, E*) which resulted in a significant elevation of the average 'baseline' $Ca^{2+}$ signal in slices compared with isolated islets (*Figure 5G*). These 'baseline' $Ca^{2+}$ pulses were glucose dependent and lowering glucose from 2.8 to 1 mM abolished all activity (*Figure 5E*).

We conclude that the $Ca^{2+}$ responses observed at 2.8 mM glucose and the shorter latency to peak $Ca^{2+}$ responses in the slices are consistent with the enhanced glucose-sensitive insulin secretion we observe (*Figure 3*) and confirm that it is the glucose-dependent trigger that is enhanced in slices. This evidence indicates increased excitability in the $Ca^{2+}$ pathway but does not suggest any mechanism that might underlie response. Furthermore, if insulin secretion was regulated by synaptic-like mechanisms then a key additional characteristic of synaptic control is that $Ca^{2+}$ channels are locally regulated presynaptically to locally deliver $Ca^{2+}$ to the sites of vesicle fusion. Interestingly, the preservation of the capillary bed in slices enabled us to determine the orientation of each β cell within the living slices and measure the $Ca^{2+}$ responses in β cells adjoining the capillary. In these cells, we often observed fast $Ca^{2+}$ waves across the cell that originated at the capillary interface (*Figure 6A–C*). This indicates a spatial clustering of functional $Ca^{2+}$ channels in the region adjoining the capillary.

In isolated islet preparations capillary structures were disrupted and observations of the $Ca^{2+}$ responses in the adjoining cells showed that $Ca^{2+}$ waves could be observed but these were rare

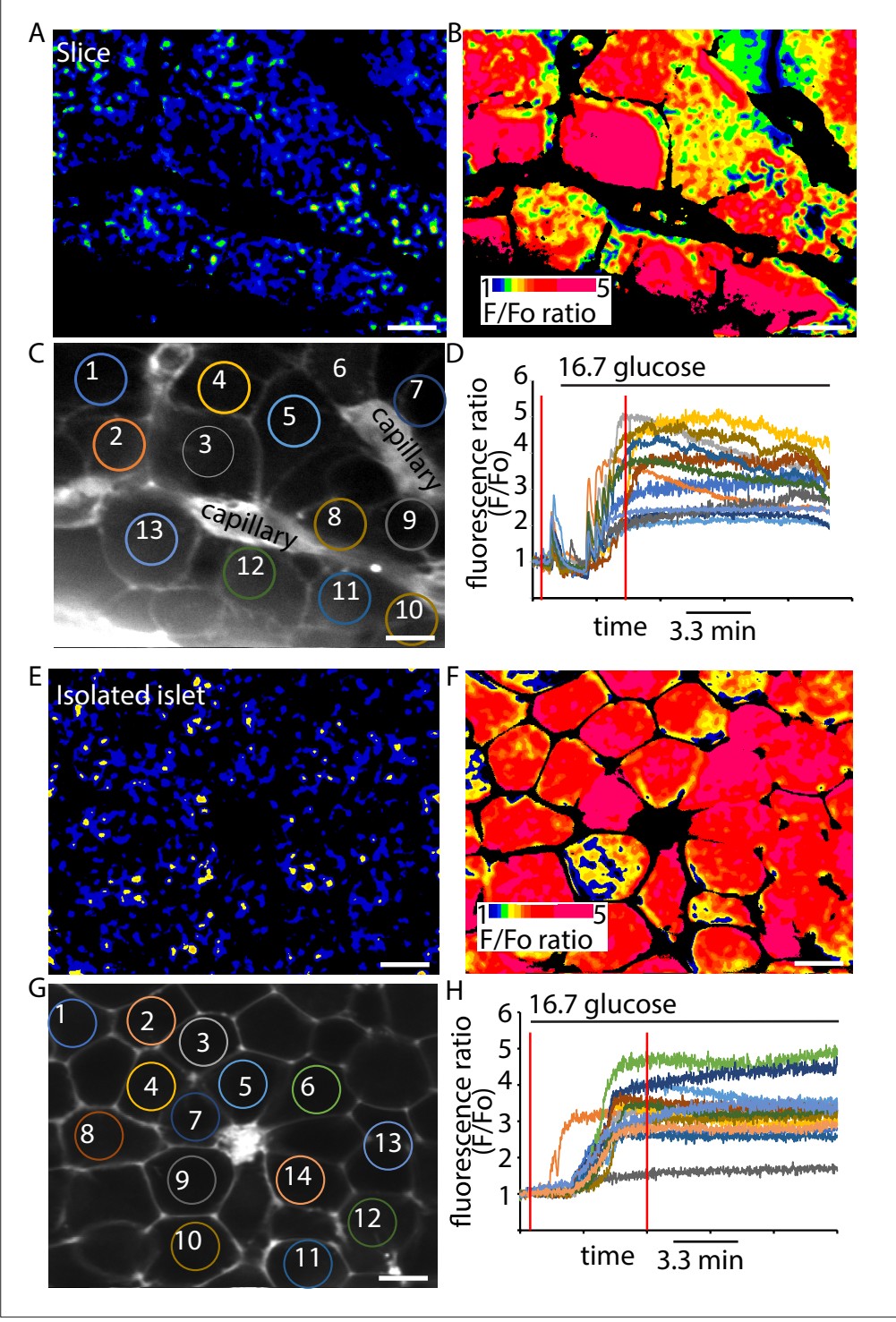

**Figure 4.** GCaMP6 recording shows synchronous glucose-induced Ca²⁺ responses in slices and isolated islets. GCaMP6 expressed in β cells shows rapid, synchronous Ca²⁺ responses in slices (**A–D**) and in isolated islets (**E–H**) in response to an increase of glucose concentration from 2.8 to 16.7 mM. Images (**F**) were ratioed against baseline fluorescence ($F_0$) before stimulation and expressed on a pseudocolour scale. (**C**) shows an image of the SRB fluorescence that outlines the cells and capillaries and identifies regions of interest of 13 cells in this example slice. (**D**) shows the trace of fluorescence ratio ($F/F_0$) plotted against time, from all 13 regions of interest. In addition, the first vertical red line shows the time point for the image (**A**) and the second line the time point for image (**B**). Glucose was added at time 0. (**G**) shows an image of the SRB fluorescence that outlines the cells and capillaries

*Figure 4 continued on next page*

*Figure 4 continued*

and identifies regions of interest of 14 cells in this example isolated islet. (**H**) shows the trace of fluorescence ratio ($F/F_0$) plotted against time, from all 14 regions of interest. In addition, the first vertical red line shows the time point for the image (**E**) and the second line the time point for image (**F**). Glucose was added at time 0. Scale bar 10 μm.

(*Figure 6D–F*). The measured velocity of Ca²⁺ waves observed for the repetitive spikes at 2.8 mM glucose was significantly faster than the waves in isolated islets (*Figure 6G, H*).

The observed Ca²⁺ waves, originating at the capillary interface, indicate mechanisms of locally increased Ca²⁺ channel activity in this region and are reminiscent of observations at the presynaptic

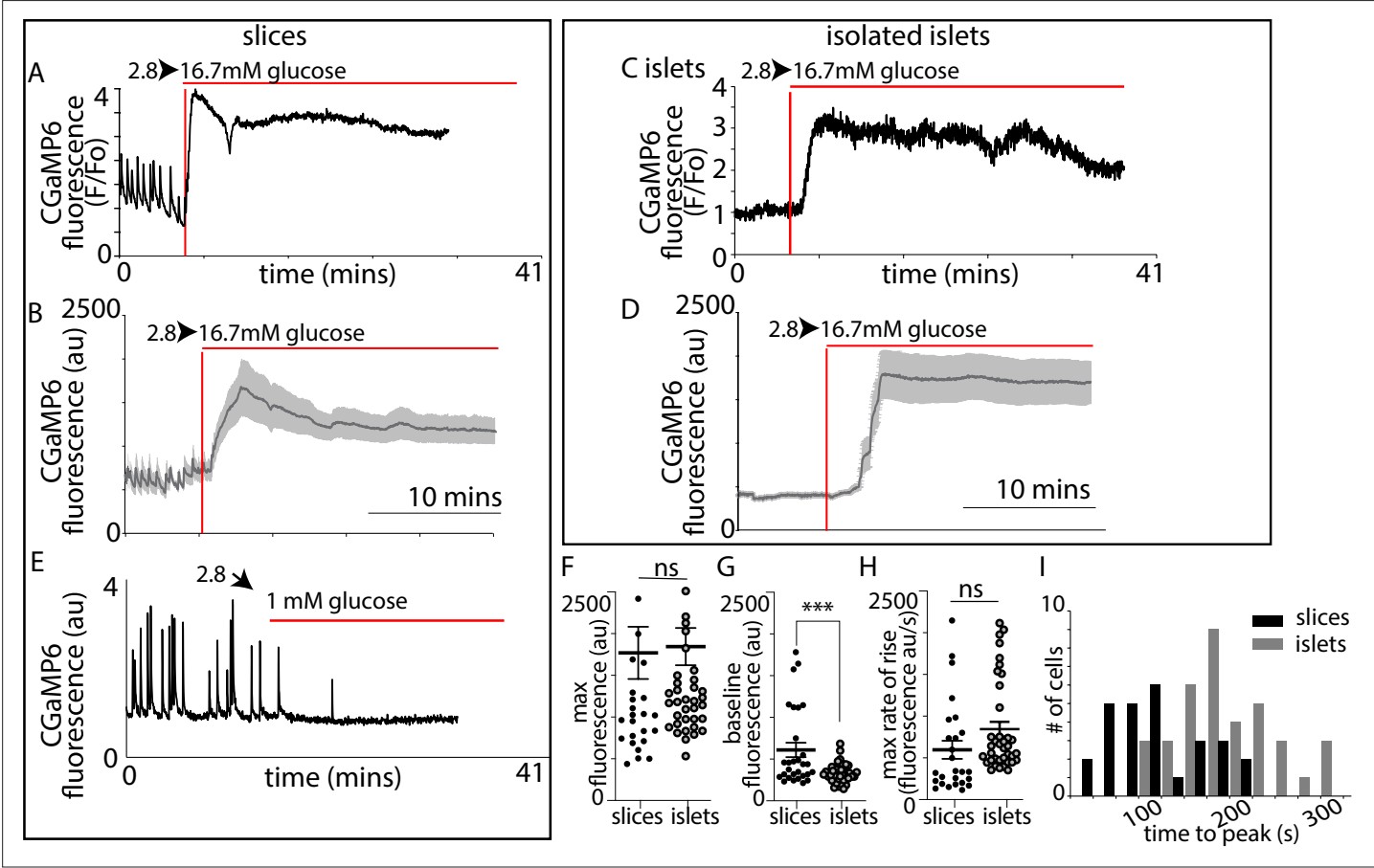

**Figure 5.** β-Cell Ca²⁺ responses in slices have short latencies to peak and higher glucose sensitivity compared to isolated islets. In slices, (**A**) single example or (**B**) averaged responses of Ca²⁺ measured by changes in GCaMP6 fluorescence in β cells within slices showed large, sustained responses to an increase of glucose from 2.8 to 16.7 mM. In slices, we often observed fast Ca²⁺ spiking in β cells (5/7 slices) prior to the increase in glucose. In isolated islets, the magnitude of (**C**) single responses, or the (**D**) average responses were similar to those in slices. (**E**) The Ca²⁺ spiking observed at 2.8 mM glucose from β cells within slices was lost when glucose was lowered to 1 mM. (**F**) The maximum fluorescence and (**H**) overall maximum rate of rise (islets *n* = 42 cells, 3 animals, in slices, *n* = 26 cells, 3 animals, Student's *t*-test p = 0.11) of the Ca²⁺ response was not different between slices and isolated islets. (**G**) In contrast, the baseline fluorescence was significantly higher in slices versus islets (in isolated islets, *n* = 43 cells, 6 islets, 3 animals and *n* = 18 cells, in slices, *n* = 18 cells, 4 slices, 3 animals, Student's *t*-test, p < 0.001). (**I**) Furthermore, the time to the peak Ca²⁺ response, using the time of arrival of glucose (with the SRB marker) was significantly delayed in islets compared with slices (178 ± 9 vs. 89 ± 10 s mean ± standard error of the mean [SEM], Student's *t*-test p < 0.01 *n* = 37 cells in islets and 27 cells in slices, *n* ≥ 3 mice) and the different preparations showed a distinct frequency time distributions. *** shows statistical significance at p<0.001.

The online version of this article includes the following source data and figure supplement(s) for figure 5:

**Source data 1.** GCaMP signal analysis.

**Source data 2.** Time to peak analysis.

**Source data 3.** Calcium wave analysis.

**Figure supplement 1.** Example record showing use of fluorescent tracer to indicate addition of high glucose.

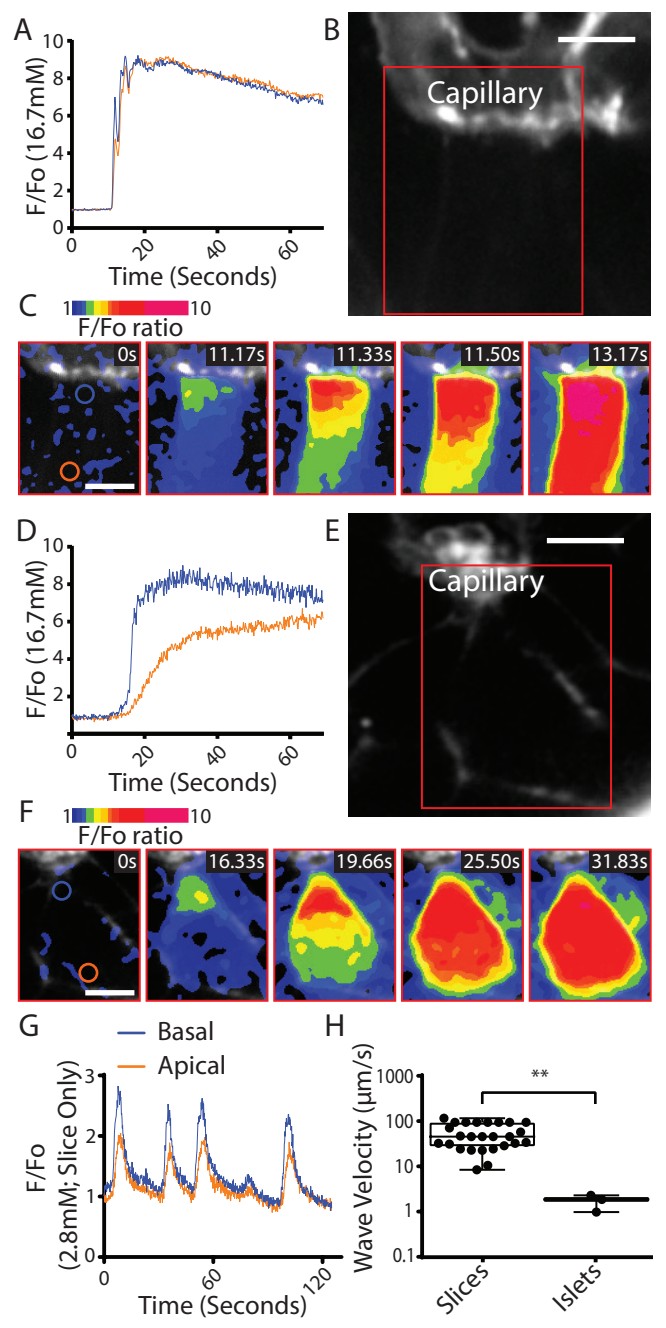

**Figure 6.** Fast $Ca^{2+}$ waves originate at the capillary interface of β cells in slices. (**A–C**) β cells within the slices that adjoin the capillaries often showed glucose (16.7 mM)-induced $Ca^{2+}$ responses that originated at the capillary interface and spread rapidly across the cell (apparent velocity 50.6 ± 6.1 µm $S^{-1}$, mean ± standard error of the mean [SEM], $n$ = 7 slices from 6 animals) to the apical region. (**B**) shows the capillary (stained with the extracellular dye SRB) the boxed region is shown in (**C**) which is a pseudocolour map of the $F/F_0$ ratio of GCaMP6 fluorescence at 5 time points over a $Ca^{2+}$ response. (**A**) shows the ratio fluorescence trace from small regions of interest (blue and red circles show in C, time 0) close to the capillary (blue trace) or distance from the capillary, the apical region (red trace). (**D–F**) In isolated islets the capillaries were fragmented, and we rarely observed $Ca^{2+}$ waves. The waves we did observe originated at the interface with capillary fragments and had a slow velocity. (1.8 ± 0.2 µm $S^{-1}$, mean ± SEM, $n$ = 3 islets from 3 animals, significantly slower compared with the velocity in slices, Student's $t$-test p < 0.01). (**E**) shows the residual capillary fragment (stained with the extracellular dye SRB) the boxed region is shown in (**F**) which is a pseudocolour map of the $F/F_0$ ratio of GCaMP6 fluorescence at five time points over a $Ca^{2+}$ response. (**D**) shows the ratio fluorescence trace from small regions of interest (blue and red circles show in F, time

*Figure 6 continued on next page*

*Figure 6 continued*

0) close to the capillary (blue trace) or distance from the capillary (red trace). (**G**) shows that during repetitive $Ca^{2+}$ spiking, recorded using GCaMP6 in slices incubated in 2.8 mM glucose, each spike shows the evidence of a $Ca^{2+}$ wave travelling from the capillary interface to the apical region. (**H**) Measurement of $Ca^{2+}$ wave velocities shows a significant reduction in islets compared to slices (Student's $t$-test $p < 0.01$, $n = 24$ waves in slices, $n = 3$ waves in isolated islets, from $n = 3$ mice) with the caveat of the scarcity of observed waves in isolated islets. Scale bar 5 μm. ** shows statistical significances at $p<0.01$.

domain. This regionally enhanced $Ca^{2+}$ channel activity is likely to be controlled by protein complexes that includes ELKS (*Ohara-Imaizumi et al., 2019*) and also by $Ca^{2+}$-dependent feedback mechanisms that are intrinsic to channel control (*Zühlke et al., 1999*). Together these mechanisms could account for the increased excitability observed in the slices and the enhanced insulin secretion.

In neurones the presynaptic complex, including $Ca^{2+}$ channels, is positioned through mechanisms that couple to the postsynaptic domain (*Südhof, 2012*). In β cells, there is no domain analogous to the postsynaptic region and therefore there must be alternative external environmental cues that position the presynaptic scaffold complex (*Lammert and Thorn, 2020*; *Ohara-Imaizumi et al., 2019*) and localise the control of the $Ca^{2+}$ channel excitability that we have revealed. We next therefore tested the most likely of these cues, the ECM and the activation of the integrin/FAK pathway which we show is preserved in the slices (*Figure 1*).

## Integrin/focal adhesion control of glucose-dependent $Ca^{2+}$ signalling

FAK phosphorylation is enhanced by glucose stimulation and the small molecular inhibitor, Y15, significantly reduces phosphorylation (*Rondas et al., 2012*). In our experiments, pretreatment of slices with Y15 completely abolished the $Ca^{2+}$ spikes observed at 2.8 mM glucose (*Figure 7A, B*) and significantly reduced the responses to 16.7 mM glucose (*Figure 7C, D*). Consistent with this inhibition, Y15 reduced glucose-induced insulin secretion in slices (*Figure 7E*) and interestingly had no effect on high potassium-induced insulin secretion. Using the granule fusion assay (shown in *Figure 2*) the cumulative number of exocytic events, in response to 16.7 mM glucose over 20 min was reduced with pretreatment with Y15 as shown when the events were mapped (*Figure 7F*) and quantified (*Figure 7G*). Furthermore, targeting of granule fusion events to the capillary interface was significantly reduced in the presence of Y15 (*Figure 7H*) supporting the idea that integrin/FAK activation localises granule fusion (*Gan et al., 2018*). Given our evidence that FAK activation is reduced in isolated islets (*Figure 1*) we tested the effect of FAK inhibition on insulin secretion in this preparation. The data show Y15 failed to inhibit insulin secretion when the islets were stimulated with high glucose or with high potassium (*Figure 7I*). This supports the idea that integrin/FAK signalling is compromised in isolated islets.

We conclude that in slices FAK is activated at the β-cell capillary interface (*Figure 1*), the same region where $Ca^{2+}$ signals originate (*Figure 6*), and that it selectively enhances glucose-dependent $Ca^{2+}$ responses. To test this idea further we moved to an in vitro model.

Culture of isolated β cells onto ECM-coated coverslips is known to enhance overall insulin secretion (*Parnaud et al., 2006*) and through local integrin activation lead to targeting of insulin granule fusion to the interface of the cells with the coverslip (*Gan et al., 2018*). But how closely this replicates the polarisation seen in native β cells within slices has not been explored.

Here, we cultured isolated β cells on laminin-coated coverslips and used immunofluorescence to determine if the structural response of the cells to contact with ECM mimicked that found in the native islet where the cells contact the ECM of the capillaries (eg *Figure 1*). The distribution of E-cadherin showed that cadherin interactions characterise cell–cell contacts (*Figure 8A, B*). Cells cultured on bovine serum albumin (BSA; as an inert protein control) did not adhere well, they grew on top of each other and although phospho-FAK was apparent at the contact points of the cells with the coverslip it was sporadic and mainly on the outer edges of the cells (*Figure 8A*). In contrast, cells cultured on laminin grew as a monolayer with extensive punctate phospho-FAK staining at the footprint (*Figure 8B*). Immunostaining for the synaptic scaffold proteins liprin and ELKS (*Figure 8C–H*) showed significant enrichment at the coverslip interface when β cells were cultured on to laminin (*Figure 8D, G, H*) and not on BSA (*Figure 8C, E, F*), which is consistent with an integrin-dependent mechanism of location both here and within slices (*Figure 1*).

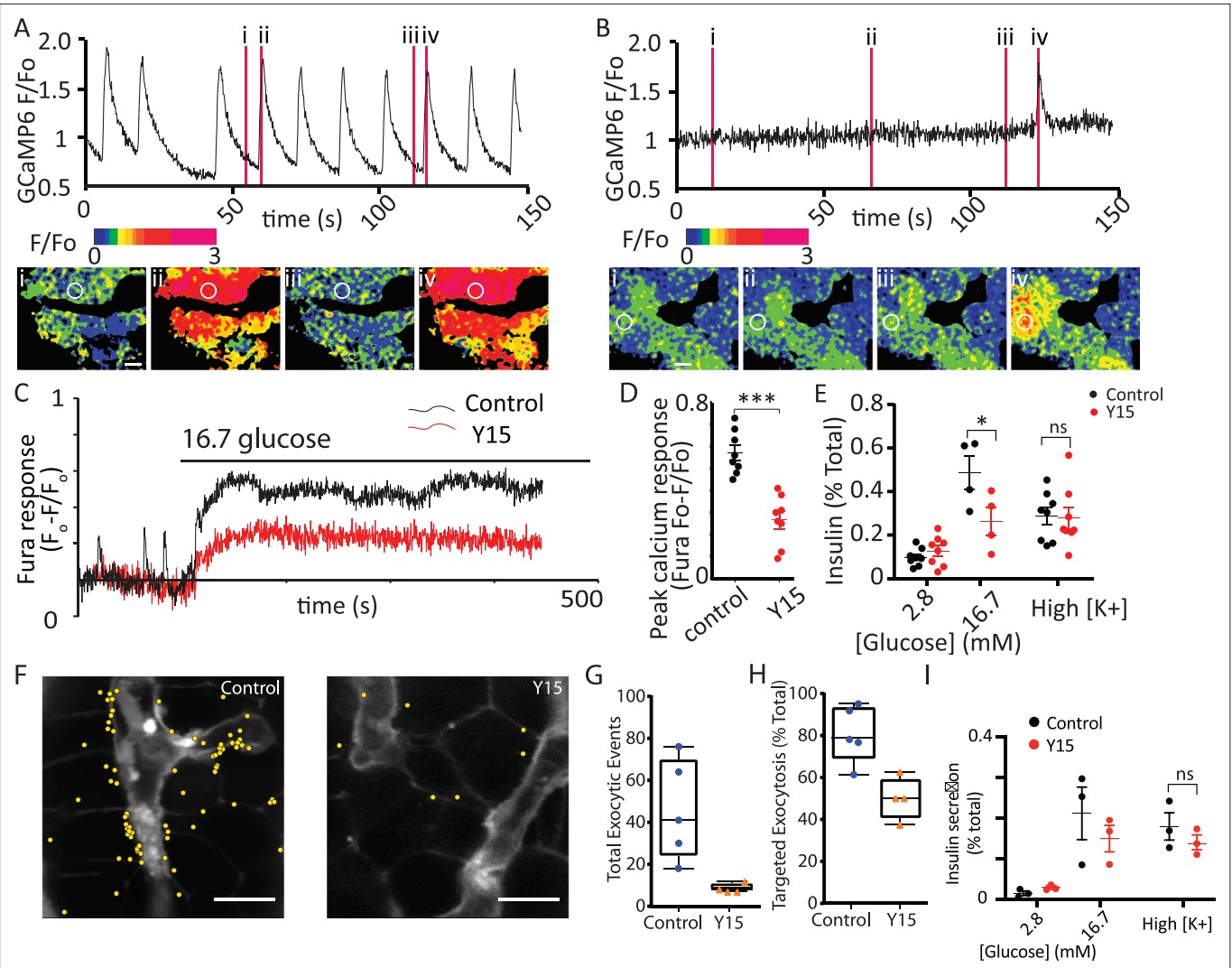

**Figure 7.** Focal adhesion kinase (FAK) activation regulates glucose-induced $Ca^{2+}$ responses. (**A**) As before, in the slice preparation, $Ca^{2+}$ spikes were observed at 2.8 mM glucose, as measured with GCaMP6 fluorescence changes. (**B**) Pretreatment of slices with 2 μM Y15, an inhibitor of FAK, blocked these $Ca^{2+}$ spikes (data from slices obtained from $n$ = 3 separate animals). (**C**) To accurately measure the peak amplitude of $Ca^{2+}$ responses we loaded cells with Fura-2, which has a lower $Ca^{2+}$ affinity than GCaMP6 ($Ca^{2+}$-induced fluorescence decreases are expressed as $F_o - F/F_o$ to normalise for the initial fluorescence and to give positive deflections with increases in $Ca^{2+}$). $Ca^{2+}$ responses to 16.7 glucose were robust in control and inhibited after pretreatment with Y15, with a significant reduction in peak amplitude (**D**, $n$ = 8 cells in slices from three separate animals, Student's $t$-test p < 0.001). (**E**) Insulin secretion measured in slices, increased in response to 16.7 mM glucose and was significantly inhibited in the presence of Y15 ($n$ = 4–8) slices from three mice, two-way analysis of variance (ANOVA) showed significant effects of Y15 ($F_{(1, 20)}$ = 6.120; p < 0.0224), glucose concentration ($F_{(1, 20)}$ = 43.82; p < 0.0001), and interaction ($F_{(1, 20)}$ = 10.36; p = 0.0043). A Tukey post hoc comparison showed Y15 significantly reduced the response at 16.7 mM glucose (p = 0.0115). Responses to high potassium (at 2.8 mM glucose) were not affected by the drug (Student's $t$-test p = 0.89, $n$ = 8 slices from three mice). (**F**) In the granule fusion assay (as described in **Figure 2**) control slices showed many exocytic events (mapped as yellow circles) in response the 16.7 mM glucose for 20 min, clustered close to the capillaries (identified as regions high in SRB). In contrast pretreatment with 2 μM Y15 decreased the number of exocytic events in response to the same glucose stimulus. (**G**) Quantification shows a reduced number of overall events after treatment with Y15 (Student's $t$-test p < 0.05, $n$ = 5 control and four slices in Y15) and (**H**) a reduced targeting of events to the capillary measured as the percentage of events within 2.5 μm of the capillary (Student's $t$-test p < 0.05, $n$ = 5 control and four slices in Y15). (**I**) Y15 applied to isolated islets had no effect on insulin secretion at either 2.8 or 16.7 mM glucose ($n$ = 3 mice, two-way ANOVA, significant main effect of glucose concentration ($F_{(1, 8)}$ = 18.94; p = 0.0024); no significant effect of Y15 ($F_{(1, 8)}$ = 0.4296; p = 0.5306) nor interaction ($F_{(1, 8)}$ = 1.102; p = 0.3245)). Y15 had no effect on high potassium-induced secretion (Student's $t$-test, p = 0.36, $n$ = 3 preparations from three mice). Scale bar 10 μm. * shows statistical significance at p<0.05; *** shows statistical significance at p<0.001.

The online version of this article includes the following source data for figure 7:

**Source data 1.** Fura calcium responses and effect of Y15.

*Figure 7 continued on next page*

*Figure 7 continued*

**Source data 2.** Secretion and effect of Y15.

**Source data 3.** Secretion and Y15.

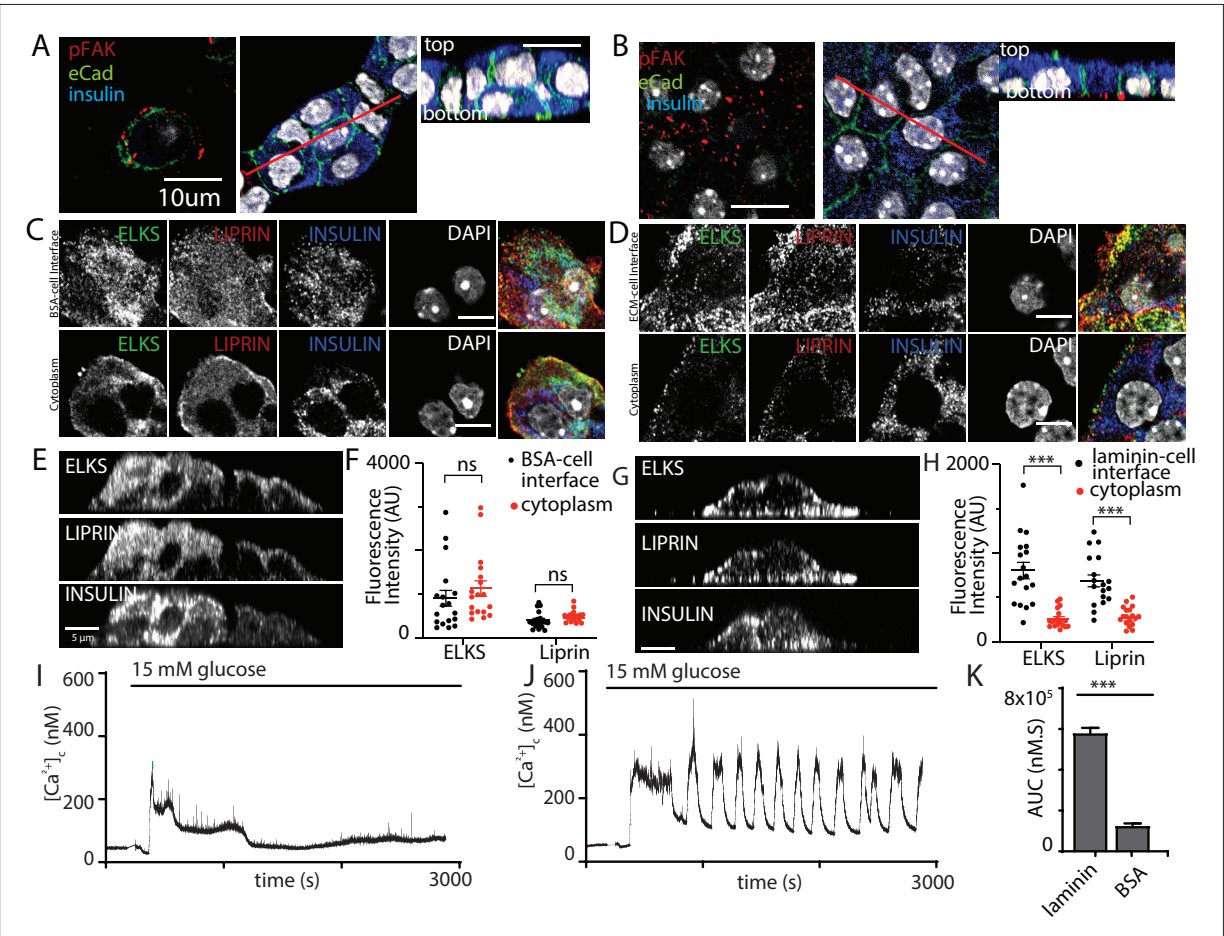

**Figure 8.** Integrin activation mediates β-cell orientation and glucose-dependent Ca²⁺ responses. (**A**) Immunofluorescence staining of phospho-FAK, E-cadherin, and insulin showed that isolated β cells, cultured on bovine serum albumin (BSA)-coated coverslips, were disorganised. Cells were multilayered and the phospho-FAK staining scattered at the edges of the footprint of the cells, also see orthogonal sections. (**B**) In contrast cells cultured on laminin-coated coverslips showed extensive, punctate phospho-FAK located at the cell footprint (as shown in the orthogonal section) and organised E-cadherin staining at the cell junctions. Immunofluorescence staining of isolated β cells (insulin; blue), grown on BSA- (**C**) or laminin- (**D**) coated coverslips showed enriched ELKS (green) and liprin (red) staining at the laminin–cell, but not BSA–cell interface, compared with the cytoplasm, see orthogonal sections (*XZ*) for cells cultured on BSA (**E**) or laminin (**G**). (**F**) Average fluorescence intensity of both ELKS (Student's *t*-test, p < 0.001) and liprin (Student's *t*-test, p < 0.05) were significantly lower at the BSA–cell interface compared with the cytosol (36 regions of interest [ROIs], *n* = 6 cells from three animals). (**H**) In the cells cultured on laminin the average fluorescence intensity of ELKS and liprin were significantly higher at the laminin–cell interface compared with the cytosol (Student's *t*-tests, p < 0.001) (36 ROIs, *n* = 6 cells from three animals). Using Fura-2-loaded, isolated β cells cultured on BSA, high glucose induced a modest, short-lasting response (**I**) that contrasted with the large response and sustained oscillations when the cells were cultured on laminin (**J**), with a significant reduction in area under the curve (AUC) of the response (K, Student's *t*-test p < 0.001, *n* = 36 cells on laminin and *n* = 21 cells on BSA). Scale bars in (**A, B**) 10 μm, all others 5 μm. *** shows statistical significance at p<0.001.

The online version of this article includes the following source data and figure supplement(s) for figure 8:

**Source data 1.** Scaffold protein distribution analysis.

**Figure supplement 1.** Blockade of integrin activation disrupts β-cell structure.

This in vitro organisation of β-cell structure therefore shares similarities with β cells in a slice including potentially a presynaptic-like domain. We therefore tested whether this would impact glucose-dependent $Ca^{2+}$ responses. β cells cultured on either BSA or on laminin showed glucose-induced $Ca^{2+}$ responses (*Figure 8I, J*) but only cells on laminin showed robust long lasting $Ca^{2+}$ oscillations and the overall AUC was significantly greater in the cells on laminin (*Figure 8K*).

This work shows that in vitro culture on laminin does not fully replicate the $Ca^{2+}$ responses seen in slices (e.g. the spikes seen at low glucose concentrations) but the comparison with cells cultured on BSA is consistent with the observed effects of FAK inhibition on $Ca^{2+}$ responses in slices (*Figure 7*). However, we were concerned that there might be non-specific effects of the different culture conditions, for example the cells on BSA grow as three-dimensional clusters. To address this, we chose acute interventions applied to β cells cultured on laminin. In the first approach, we pretreated the

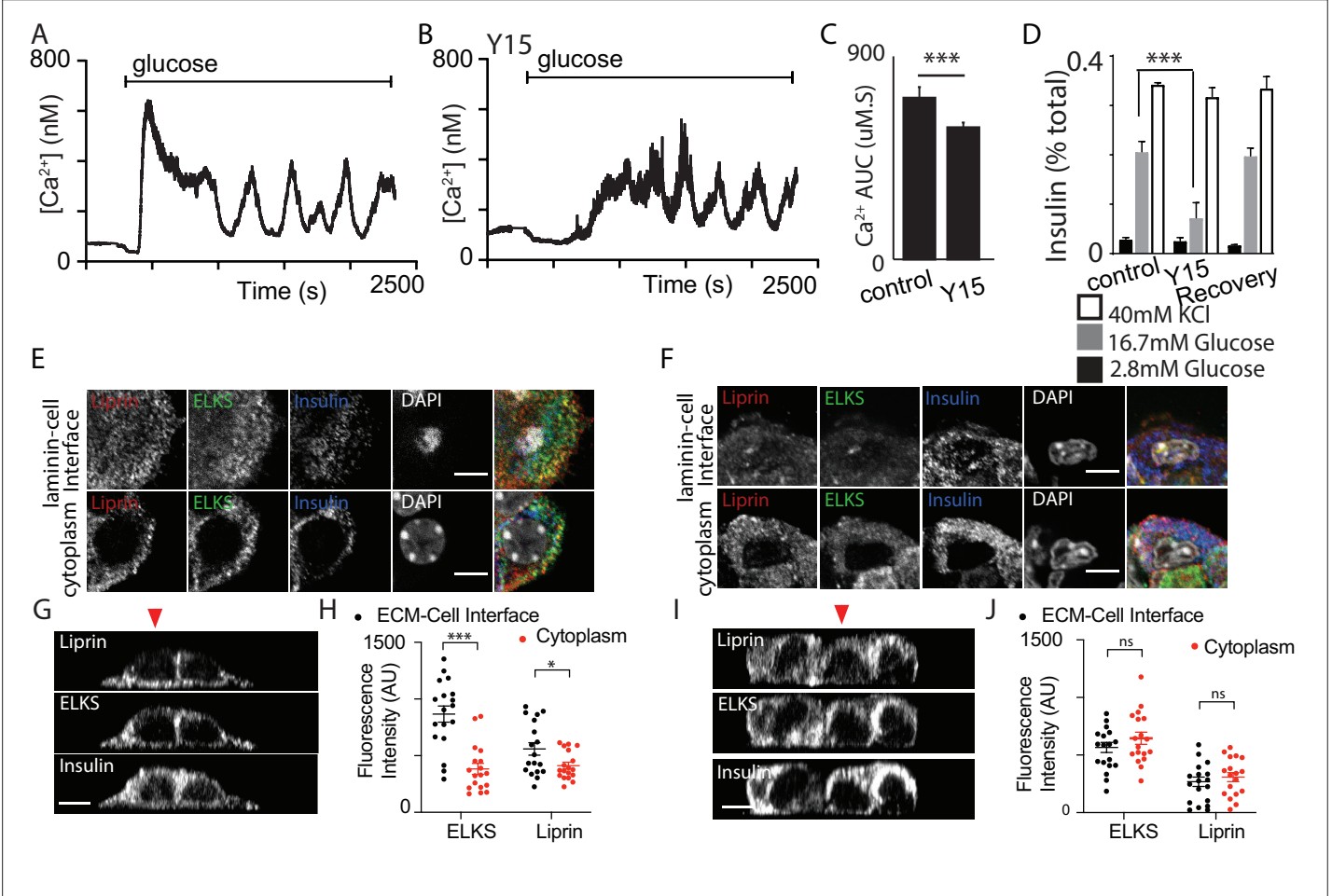

**Figure 9.** Focal adhesion kinase (FAK) regulates both $Ca^{2+}$ responses and positioning of presynaptic scaffold proteins. In Fura-2-loaded cells (cultured on laminin), we observed the typical robust response to high glucose followed by sustained oscillations in control (**A**). A smaller, delayed response was observed in the presence of 2 μM Y15 (**B**) with a significant reduction in AUC (C, using regions of interest [ROIs] from n = 218 cells in DMSO and 208 cells in Y15, measured over the total time of 2500 s, from three mice, Student's *t*-test p < 0.001). (**D**) Insulin secretion, measured in dispersed cells cultured on laminin, was reversibly (after a 6 hr wash) reduced in the presence of Y15 (n = 3 animals in each condition, Student's *t*-test p < 0.001). No significant difference in insulin secretion was observed following potassium stimulation between cells incubated with Y15 compared with DMSO control (n = 3 animals, Student's *t*-test p = 0.25). (**E, G, H**) As before, immunostaining showed enrichment of liprin and ELKS at the laminin–cell interface which was blocked after pretreatment with Y15 (F, I, J, ELKS Student's *t*-test, p = 0.15 and liprin, Student's *t*-test, p = 0.28, 36 ROIs, n = 6 cells from three animals). Scale bar 5 μm. * shows statistical significance at p<0.05; *** shows statistical significance at p<0.001.

The online version of this article includes the following source data for figure 9:

**Source data 1.** Effect of Y15 on calcium responses.

**Source data 2.** Insulin secretion and effect of Y15.

**Source data 3.** Scaffold protein distribution and Y15.

cultures with integrin-β1 function-blocking antibodies (*Mendrick and Kelly, 1993*) and, consistent with the data in *Figure 7* we saw both a disruption in the localisation of liprin at the coverslip interface and an inhibition of the glucose-induced Ca$^{2+}$ responses (*Figure 8—figure supplement 1*).

In the second approach, we used the FAK inhibitor Y15 applied to β cells cultured on laminin-coated coverslips. In the presence of Y15, the glucose-induced Ca$^{2+}$ response was significantly reduced (*Figure 9A–C*) and glucose-induced secretion, but not high potassium, was also inhibited in a reversible manner (*Figure 9D*), both consistent with the actions of Y15 in the slices (*Figure 7*). Immunofluorescence studies showed that the distribution of liprin and ELKS were disrupted by Y15 (*Figure 9E–J*), consistent with the data showing the importance of the integrin/FAK pathway in their positioning.

Taken together our data provide strong evidence that the integrin/FAK pathway is critical both for the local enrichment of synaptic scaffold proteins in β cells and for locally shaping the Ca$^{2+}$ responses.

## Discussion

Our interrogation of β-cell structure and function in pancreatic slices shows precise subcellular organisation, targeting of granule fusion to the capillary interface and enhanced insulin secretion that points to a robust glucose-dependent trigger. We observe Ca$^{2+}$ spikes at low glucose and short-latency responses to high glucose showing enhanced sensitivity of the cells to glucose in slices compared to isolated islets. Using a range of interventions, we show a glucose-dependent integrin/FAK pathway locally enhances the Ca$^{2+}$ response and positions the presynaptic scaffold proteins, ELKS and liprin. This work demonstrates that the FAK pathway intersects with the final stages of the glucose-dependent control of secretion and has important implications for our understanding of the stimulus secretion cascade in β cells and treatments for diabetes.

### FAK and the control of insulin secretion

We are not the first to identify a role for FAK in the control of secretion. Halban's group showed in mouse β cells that FAK phosphorylation was increased by glucose stimulation (*Rondas et al., 2011*) and block of integrins or FAK inhibited insulin secretion from the MIN6 cell line with evidence that it affected F-actin remodelling (*Rondas et al., 2012*). In a mouse study, knockout of FAK caused hyperglycemia and using isolated islets they showed a reduced insulin secretion but no effect on Ca$^{2+}$ responses (*Cai et al., 2012*). However, our work now shows that we must be careful in interpreting data from isolated islets. The dramatic reduction in phospho-FAK compared to slices (*Figure 1*) means the integrin/FAK pathway is compromised in isolated islets, something we directly show (*Figure 7J*). Interestingly, Halban's approach cultured the β cells onto dishes coated with ECM (*Rondas et al., 2011*) which, since we now demonstrate is an excellent model that recapitulates FAK activation, β-cell organisation, Ca$^{2+}$ signals and secretory responses, is a much better approach to explore this pathway.

These previous studies did not explore the subcellular actions of the integrin/FAK pathway and, although they imply an action on F-actin, the mechanism was not identified. In contrast, we show direct evidence that FAK is a master regulator of two processes in the latter stages of glucose-dependent control of insulin secretion where it controls both the positioning of presynaptic scaffold proteins and the Ca$^{2+}$ signal.

### Evidence that the integrin/FAK pathway regulates synaptic-like mechanisms to control insulin secretion

In neurones, the key steps from opening of voltage-gated Ca$^{2+}$ channels to the exocytic fusion of vesicles are tightly regulated by presynaptic complexes (*Südhof, 2012*). In β cells, closely analogous steps use glucose-dependent Ca$^{2+}$ signals to induce insulin granule fusion, furthermore, presynaptic scaffold proteins are present (*Low et al., 2014*; *Ohara-Imaizumi et al., 2005*) and function to control insulin secretion (*Fujimoto et al., 2002*; *Ohara-Imaizumi et al., 2005*; *Shibasaki et al., 2004*). However, whether these scaffold proteins exist as a complex that regulate insulin secretion in a manner analogous to synaptic control is not clear.

Here, we provide evidence that aspects of the control of insulin secretion in β cells are similar to presynaptic mechanisms. We show that presynaptic scaffold proteins, insulin granule fusion and the control of Ca$^{2+}$ channels all occur locally where the β-cell contact ECM. Furthermore, activation of the

integrin/FAK pathway is critical for each one of these factors, either in positioning of granule fusion as we have previously shown (*Gan et al., 2018*) or, as we now show in the positioning of the scaffold proteins and regulation of the $Ca^{2+}$ response.

In terms of spatial constraints, liprin, ELKS, and other presynaptic scaffold proteins are all enriched at the capillary interface (*Figure 1*; *Low et al., 2014*) and when this complex is preserved, as we now show in slices, there is a very tight focus of insulin granule fusion to this region (*Figure 2*). This is consistent with a synaptic-like mechanism. The various roles of liprin in neurones are still being uncovered but through protein–protein interactions it nucleates the formation of the presynaptic complex including proteins such as RIM which in turn tether granules (*Südhof, 2012*; *Wei et al., 2011*). Future work will be required to identify if liprin plays a similar role in β cells.

We emphasise here that an advance of our new work is the ability to record subcellular responses from individual β cells within slices. Previous work has exploited fixed slices for 3D confocal microscopy (*Cottle et al., 2021*; *Gan et al., 2017*; *Low et al., 2014*) but functional subcellular imaging has been difficult and, for example, recording of targeting of insulin granule fusion (*Low et al., 2014*) or the spatial complexity of $Ca^{2+}$ responses (*Ohara-Imaizumi et al., 2019*) has been performed using isolated islets. The work presented here not only advances our understanding of functional responses in β cells within slices but allows a direct comparison with the structure and function of isolated islets under identical conditions.

In terms of control of the $Ca^{2+}$ response, our new evidence indicates that synaptic-like mechanisms play a role. The $Ca^{2+}$ responses we observe are a spatial and temporal integration of discrete bursts of $Ca^{2+}$ entry at each action potential (*Rorsman and Ashcroft, 2018*). Our data show that the maximal global rate of rise of the GCaMP measured $Ca^{2+}$ response is similar between the slices and isolated islets (*Figure 5H*). This suggests that the number of active $Ca^{2+}$ channels in the β cells in both preparations is similar and is therefore consistent with the long-standing observations of robust $Ca^{2+}$ responses in isolated islets. What is different in the slices is that we observe rapid local increases in $Ca^{2+}$ and waves at the capillary interface, which must reflect local clustering of active channels – a central characteristic of neuronal synapses.

How do we explain the enhanced sensitivity to glucose of the $Ca^{2+}$ responses in slices? Specifically, we might expect mechanisms that act on the voltage sensitivity of the $Ca^{2+}$ channels, so they respond at more negative membrane potentials, or that the $Ca^{2+}$ channels open longer and increase $Ca^{2+}$ influx. Our data provide evidence for two possible factors that are shaping the $Ca^{2+}$ responses in slices. Firstly, the clustering of active $Ca^{2+}$ channels at the capillary interface will affect $Ca^{2+}$ channel behaviour. The mouse has a diversity of CaV channels (*Yang and Berggren, 2005*) but evidence shows that Cav1.2 plays a major role in the control of secretion (*Schulla et al., 2003*) and is positively and negatively regulated by cytosolic $Ca^{2+}$ (*Zühlke et al., 1999*). As has been shown in many other systems, the entry of $Ca^{2+}$ through each channel influences its own activity and the activity of immediately surrounding channels which makes channel clustering a critical factor in controlling channel opening (*Stanley, 1997*). Secondly, the localised activation of focal adhesions (*Figure 1*) may target the regulation of $Ca^{2+}$ channels. We show that culture of cells on BSA, inhibition of FAK and integrin-β1 blockade all reduce the $Ca^{2+}$ response to glucose. This is the first report of a link between integrins and $Ca^{2+}$ response in β cells, which could be mediated through signal cascades elicited by focal adhesion activation, as has been shown in smooth muscle cells (*Hu et al., 1998*) or it could be secondary to an integrin/FAK-mediated positioning of synaptic scaffold proteins. For the latter, we have shown integrin activation positions liprin and ELKS (*Figure 8*) and in turn ELKS may position the $Ca^{2+}$ channels (*Ohara-Imaizumi et al., 2019*).

## How might the presynaptic-like complex be positioned at the capillary interface in β cells?

One point of distinction in the β cell compared to neurones is that there is no equivalent to a post-synaptic domain. In neurones, the pre- and post-synaptic domains are aligned by transmembrane proteins that span the synaptic cleft, such as neurexins (*Südhof, 2008*). Indeed, neurexins do exist in β cells (*Mosedale et al., 2012*) but our work now suggests that the integrin/focal adhesion pathway is a more likely candidate controlling the positioning of the presynaptic complex and we directly show it controls the positioning of both ELKS and liprin.

The question arises as to how this occurs and although there is evidence that liprins do interact with focal adhesions (*Astro et al., 2016*) this has not been explored in β cells. Emerging new data is pointing towards a role for cortical complexes that contain ELKS and liprin and regionally locate microtubule plus ends to the subplasmalemmal regions (*Grimaldi et al., 2014*; *Lansbergen et al., 2006*). What is interesting is that these cortical complexes are now shown to link with focal adhesions through KANK1 (*Bouchet et al., 2016*) and many components of these complexes have recently been identified in β cells (*Noordstra et al., 2022*). Clearly further work is needed to understand how this system might function and control insulin secretion, but it is an attractive, and testable model that localises microtubules and presynaptic scaffold proteins with cortical complexes and focal adhesions all to the capillary interface of β cells.

## Enhanced sensitivity to glucose in slices

Our finding of enhanced sensitivity to glucose in the pancreatic slices is a significant advance in the field. We observe repetitive $Ca^{2+}$ spikes at 2.8 mM glucose that are lost when glucose is lowered to 1 mM and are not seen in isolated islets. In parallel, insulin secretion is observed from slices at 2.8 mM and decreases when glucose is lowered. This enhanced glucose sensitivity is likely to be driven by the intrinsic factors within the β cells we have identified. These factors include the identification of fast $Ca^{2+}$ waves that originate at the capillary interface, the short latency to peak $Ca^{2+}$ responses and the close coupling between the $Ca^{2+}$ signals and sites of insulin granule exocytosis. We cannot rule out that other factors, present in pancreatic slices, may influence glucose sensitivity. One possible factor is the gap junction coupling of the cells, where, at low glucose concentrations, a majority of non-responsive cells are thought to supress the activity of individual particularly excitable cells (*Benninger et al., 2011*). However, this does not seem a likely explanation for our findings because we observe strong coordination of $Ca^{2+}$ responses, indicative of cell-to-cell coupling, in both slices and isolated islets (*Figure 3*). Another obvious factor, that might differ in the preparations, is α cells where glucagon secretion can stimulate insulin release (*Moede et al., 2020*). However, this seems unlikely because lowering glucose from 2.8 to 1 mM would stimulate glucagon secretion and in the β cells we observe the opposite; a reduced insulin secretion and a reduced $Ca^{2+}$ response.

In a broader physiological context, it might seem unlikely that the responses we observe to low glucose concentrations are real. The 'set point' for mouse blood glucose is ~7 mM (*Rodriguez-Diaz et al., 2018*) and the consensus from other studies, mostly using isolated islets, is that insulin secretion has an $EC_{50}$ for glucose of ~8 mM (*Hedeskov, 1980*). Furthermore, the $K_m$ for the GLUT 2 transporter is 11 mM and the $EC_{50}$ for mouse glucokinase is 8 mM (*Rorsman and Ashcroft, 2018*). The $EC_{50}$ we observe in slices is 8.6 mM and so is consistent with this past work and with physiological relevance, but the key distinction in our findings is the much greater sensitivity to lower glucose concentrations. There is however precedent that β cells can respond to much lower glucose concentrations. Using in vitro approaches Henquin's lab showed a dose dependence of the amplifying pathway from 1 to 6 mM (*Gembal et al., 1992*) and extensive early work identified subpopulations of isolated β cells that are very sensitive to glucose and released maximal insulin at 8.3 mM glucose (*Van Schravendijk et al., 1992*), similar to our findings (*Figure 3*). More recently, using a perfused pancreas preparation, a significant increase in insulin secretion was observed at 6 mM glucose compared to 1 mM glucose (*Kellard et al., 2020*). Given the excellent preservation of cell structure within the slice, our results likely reflect optimal behaviour of β cells and this could underpin their responsiveness to low glucose.

In vivo the control of glucose is dependent on a balance of hormones (*Rodriguez-Diaz et al., 2018*) with evidence, in primates and humans that even in a fasting state, insulin, glucagon, and blood glucose concentrations synchronously oscillate indicating that hormone secretion is never zero (*Goodner et al., 1977*; *Song et al., 2000*). It is interesting that insulin secretion at these low glucose concentrations is pulsatile since this is consistent with the $Ca^{2+}$ spikes we observe. We conclude that our observations of insulin secretion in slices are interesting and likely to be a reflection of the better preservation of β-cell architecture and altered $Ca^{2+}$ responses. However, we cannot rule out the possibility that other factors in the slices are involved and more work will be required to determine how our findings relate to native regulation of β-cell function.

## Broader significance

Our work has important implications for understanding and treating diabetes. We have recently shown that the fundamental relationships between β cells and capillaries are similar between mouse and human islets (*Cottle et al., 2021*), suggesting that the integrin/FAK pathway may play a similar role in human β cells. For type 2 diabetes, past work has indicated an impact of lipotoxicity on $Ca^{2+}$ channel organisation (*Hoppa et al., 2009*) and the disease on $Ca^{2+}$ clustering (*Gandasi et al., 2017*) which, in the new context given by our work, would take place at the capillary interface. We also know that both the capillary structure (*Brissova et al., 2015*) and the ECM composition (*Hayden et al., 2005*) are altered in disease. Furthermore, it is shown that β-cell function is compromised, as type 2 diabetes develops, prior to loss of β-cell mass (*Cohrs et al., 2020*). In the light of our work, it is possible that these functional changes to β-cell responses might result from disruption of the capillaries and effects on the integrin/FAK pathway we describe. Given that sulfonylureas can improve insulin secretion in T2D (*Rorsman and Ashcroft, 2018*) we already know that enhancement of glucose-dependent triggering is beneficial. Our new work suggests that widening the scope of our interest to include each element of the triggering pathway would be fruitful and that specifically intervening with the primary mechanisms that spatially organise the β cells could be disease modifying.

For type 1 diabetes, exciting advances are leading to the development of stem cell-based β-cell replacements (*Melton, 2021*). Most approaches generate spheroids of cells that we have recently shown do not contain organised ECM (*Singh et al., 2021*) and, as a result, the β-like cells within the spheroids are not polarised (*Singh et al., 2021*). Our work now suggests that amplification will be the dominant pathway underpinning glucose-dependent insulin secretion in these spheroids and that these cells will lack a drive from the integrin/FAK pathway. Because the triggering and amplification pathways are distinct our work indicates that a selective focus on enhancement of triggering may be broadly beneficial. This could include imposing polarity to the β-like cells, which we have shown does enhance secretion (*Singh et al., 2021*), but it could also include genetic manipulation to upregulate components of the triggering pathway or the use of drugs, like sulfonylureas, to increase the sensitivity of this pathway.

In summary, our work exploits the pancreatic slice technique to highlight the importance of β-cell architecture and the islet environment in controlling glucose-dependent insulin secretion. Ongoing work is needed to determine the role of synaptic-like control of insulin secretion in healthy and diseased β cells.

## Materials and methods

### Animal husbandry

Male C57BL/6 and GCAMP-InsCre mice were produced from mouse *Ins1Cre* (The Jackson Laboratory, strain #: 026801 B6(Cg)-Ins1^tm1.1(cre)Thor/J) mice crossed with GCaMP6s mice (The Jackson Laboratory, strain #: 024106 B6;129S6-Gt(ROSA)26Sortm96(CAG-GCaMP6s)Hze/J) housed at the Charles Perkins Centre facility in a specific pathogen-free environment, at 22°C with 12-hr light cycles. All mice were fed a standard chow diet (7% simple sugars, 3% fat, 50% polysaccharide, 15% protein (wt/wt), energy 3.5 kcal/g). Mice (8–12 weeks old) were humanely killed according to local animal ethics procedures (approved by the University of Sydney Ethics Committee).

### Glucose-stimulated insulin secretion and Homogeneous Time Resolved Fluorescence insulin assay

Glucose-stimulated insulin secretion (GSIS) media was Krebs–Ringer bicarbonate solution of pH 7.4 buffered with N-(2-Hydroxyethyl)piperazine-N'-(2-ethanesulfonic acid) (HEPES) (KRBH), plus 2.8 mM glucose (basal) or 16.7 mM glucose (stimulation) composed of: 120 mM NaCl, 4.56 mM KCl, 1.2 mM $KH_2PO_4$, 1.2 mM $MgSO_4$, 15 mM $NaHCO_3$, 10 mM HEPES, 2.5 mM $CaCl_2$, and 0.2% BSA, pH 7.4. This media was used in all the insulin measurements from slices, isolated islets and dispersed cells.

Depolarisation media was a modified KRBH with reduced NaCl (100 mM) and high potassium (40 mM KCl). Where applied, diazoxide (Sigma) was used at a concentration of 250 µM. All media and cells were kept at 37°C for the duration of the assay. Tissues were washed in warm basal media two times and then placed in fresh basal media for 1 hr. The basal media was washed out an additional time and then tissues were incubated for 30 min in fresh basal media. Tissues were collected at the

end of the assay into ice-cold lysis buffer (1% NP-40, 300 mM NaCl, 50 mM Tris–HCl pH 7.4, protease inhibitors) and sonicated. Supernatants and lysates were stored at −30°C prior to HTRF assay (Mouse ultrasensitive, Cisbio).

## Islet preparation

Isolated mouse islets were prepared according to a standard method that utilises collagenase enzymes for digestion and separation from exocrine pancreatic tissue (*Hoppa et al., 2009*). In brief, a Liberase (TL Research grade, Roche) solution was prepared in unsupplemented RPMI-1640 (Gibco) media at a concentration of 0.5 U/ml. Pancreases were distended by injection of 2 ml of ice cold Liberase solution via the pancreatic duct, dissected and placed into sterile tubes in a 37°C shaking water bath for 15 min. Isolated islets were separated from the cell debris using a Histopaque (Sigma) density gradient. Isolated islets were maintained (37°C, 95/5% air/CO$_2$) in RPMI-1640 culture medium (Sigma-Aldrich), 10.7 mM glucose, supplemented with 10% fetal bovine serum (FBS; Gibco, Victoria, Australia), and 100 U/ml penicillin/0.1 mg/ml streptomycin (Invitrogen, Victoria, Australia).

## Islet slices

Sectioning of unfixed pancreatic tissue was performed as described by Huang et al. (*Gan et al., 2017*; *Huang et al., 2011*). Pancreatic sections (200 μm thick) were cut and incubated overnight in RPMI-1640 supplemented with penicillin–streptomycin, 10% FBS, and 100 μg/ml soybean trypsin inhibitor (Sapphire Bioscience).

## Tissue fixation and immunofluorescent staining

Tissues were fixed with 4% paraformaldehyde (Sigma-Aldrich) in phosphate-buffered saline (PBS) 15 min at 20°C. Samples were stored in PBS at 4°C prior to immunofluorescent staining. Immunofluorescence was performed as described by Meneghel-Rozzo et al. (*Meneghel-Rozzo et al., 2004*). Tissues were incubated in blocking buffer (3% BSA, 3% donkey serum, 0.3% Triton X-100) for a minimum of 1 hr at room temperature followed by primary antibody incubation at 4°C overnight. Sections were washed in PBS (4 changes over 30 min) and secondary antibodies (in block buffer) were added for 4 hr (whole islets and slices) or 45 min (cells) at 20°C. After washing in PBS, tissues were mounted in Prolong Diamond anti-fade reagent (Invitrogen).

## Imaging

Confocal imaging was performed on a Nikon C2 microscope using a ×63 oil immersion objective or on a Leica SP8 microscope with a ×100 oil immersion objective. Live-cell imaging was possible on a two-photon microscope constructed in-house using Olympus microscope components. Two-photon imaging was performed at 37°C. Images were analysed using ImageJ and MetaMorph software. A 3D circumference linescan analysis (for example in *Figure 1J, P*) used linescans around the cell circumference at each *Z* section. The fluorescence intensity along each circumference linescan was then plotted out as intensity plots to produce the 3D heatmaps. The heatmap was produced in Excel by assigning pseudocolours to fluorescence intensity. Quantitation of protein area (*Figure 1C, D*) was calculated by converting single channels to binary images using a threshold that eliminated background (estimated as the average signal in the area of the nucleus) and was normalised to total cell area as measured by the combined area of insulin expression and DAPI expression.

## Islet slices and Fura-2 measurement

Slices were removed from overnight culture media and incubated in 6-well plates containing 1 ml KRBH 11 mM glucose with 6 μm Fura 2-AM, 2 slices per well on a rocking platform at room temperature for 1 hr. After incubation, slices were placed back in culture media and washed for up to 6 hr in an incubator set to 37°C and 5% CO$_2$. Slices were removed for experimentation as needed and imaged after a pre-basal period of 1 hr in KRBH 2.8 mM glucose with or without the presence of 2 μm Y15 in an incubator. After pre-basal, single slices were removed and placed in a pre-heated imaging chamber at 37°C with 1 ml KRBH 2.8 mM glucose. Slices were stimulated by adding glucose solution to a final concentration of 16.7 mM and imaged with an excitation laser tuned to 810 nm on a two-photon microscope and emitted light collected between 470 and 520 nm. Sulforhodamine B was used at a concentration of 400 μm to visualise capillaries and recorded in a separate channel >590 nm.

## Antibodies

Primary antibodies used for this study were: anti-insulin (Dako Cytomation, A0564), anti-beta1 laminin (Thermo Scientific MA5-14657), anti-integrin beta 1 (BD Biosciences 555002), anti-talin (Sigma-Aldrich T3287), anti-phosphorylated FAK (Cell Signalling Tech 8,556S), anti-liprin alpha1 (Proteintech 14175-1-AP), and anti-ELKS (Sigma, E4531). All primary antibodies were diluted 1/200. Secondary antibodies were highly cross absorbed donkey or goat antibodies (Invitrogen) labelled with Alexa 488, Alexa 546, Alexa 594, or Alexa 647. All were used at a 1/200 dilution. DAPI (Sigma, 100 ng/ml final concentration) was added during the secondary antibody incubation.

| Target | Species | Manufacturer/catalogue number |
|---|---|---|
| Insulin | Guinea pig | DAKO, AO564 |
| Laminin-beta1 | Rat | Invitrogen, MA5-14657 |
| Integrin-beta1 | Hamster | BD Biosciences, 555,002 |
| Liprin-alpha1 | Rabbit | Proteintech, 14175-1-AP |
| PAR3 | Rabbit | Millipore, 07-330 |
| E-cadherin | Mouse | BD Biosciences, 610,181 |
| Phospho-FAK (Y397) | Rabbit | CST, 8556S |

## Islet cell seeding procedure

Single-cell suspensions were prepared by digesting isolated islets with TrypLE express enzyme (Gibco). Culture medium was RPMI-1640 supplemented with 10% FBS, and 100 U/ml penicillin/0.1 mg/ml streptomycin. Cells were cultured in standard incubator conditions (37°C, 10% $CO_2$, humidity 20%).

In most experiments (*Figure 8*), we simply used plain coverslips but in the insulin secretion assays (*Figure 9D*), to create a more stable covalent attachment of basement membrane proteins to the surface of the glass coverslips we coated the coverslips with a thin layer (approximately 10–20 nm thick) of plasma activated coating (see *Kosobrodova et al., 2018* for details). The plasma treatment was conducted using a radio frequency (RF) power supply (Eni OEM-6) powered at 13.56 MHz and equipped with a matching box. Plasma ions were accelerated by the application of negative bias pulses from RUP6 pulse generator (GBS Elektronik GmbH, Dresden, Germany) for 20-μs duration at a frequency of 3000 Hz to the stainless-steel sample holder. Glass coverslips were first activated in argon plasma powered at 75 W under a 500 V negative bias for 10 min at 80 mTorr. After that, a gas flow consisting of acetylene (1 sccm), nitrogen (3 sccm), and argon (13 sccm) was introduced into the chamber for 10-min plasma deposition. During this step, plasma was generated with 50 W RF power at a pressure of 110 mTorr while positive ions were deposited on glass coverslips under a negative bias of 500 V. After the plasma treatment, activated coverslips were kept in a petri dish in ambient conditions until use.

Plasma-treated coverslips or plain coverslips were coated with Laminin 511 (BioLamina) 5 μg/ml or BSA (Sigma) 1 mg/ml overnight at 4°C. After coating coverslips were rinsed in PBS and then the cells were seeded.

## Statistical analyses

All numerical data are presented as mean ± standard error of the mean. Statistical analysis was performed using Microsoft Excel and GraphPad Prism. Datasets with two groups were subjected Student's *t*-test, unpaired, equal variance. Analysis of variance was applied to experiments with multiple parameters, one- or two-way as appropriate. And, where required, significance analysed using a post hoc Tukey test. Significance is indicated as follows: *$p < 0.05$, **$p < 0.01$, ***$p < 0.001$.

## Acknowledgements

We acknowledge project funding obtained from the National Health and Medical Research Council (APP1128273, to PT), The University of Sydney Strategic Research Excellence Initiative (SREI to PT and

MB), Diabetes Australia (DART grant to PT), and Australian Research Council (FL190100216 to MB). Imaging was performed in the Centre for Microscopy and Microanalysis at the University of Sydney.

## Additional information

### Funding

| Funder | Grant reference number | Author |
|---|---|---|
| National Health and Medical Research Council | APP1128273 | Peter Thorn |
| Sydney Medical School | SREI | Peter Thorn Marcela Bilek |
| Diabetes Australia Research Trust | | Peter Thorn |
| Australian Research Council | FL190100216 | Marcela Bilek |

The funders had no role in study design, data collection, and interpretation, or the decision to submit the work for publication.

### Author contributions

Dillon Jevon, Kylie Deng, Conceptualization, Formal analysis, Investigation, Methodology, Writing – review and editing; Nicole Hallahan, Conceptualization, Formal analysis, Investigation, Methodology, Supervision, Writing – original draft, Writing – review and editing; Krish Kumar, Clara Tran, Investigation, Methodology; Jason Tong, Wan Jun Gan, Conceptualization, Investigation, Methodology; Marcela Bilek, Investigation, Methodology, Supervision; Peter Thorn, Conceptualization, Data curation, Formal analysis, Funding acquisition, Investigation, Methodology, Project administration, Resources, Supervision, Writing – original draft, Writing – review and editing

### Author ORCIDs

Kylie Deng http://orcid.org/0000-0002-9096-2574
Jason Tong http://orcid.org/0000-0003-1027-3662
Peter Thorn http://orcid.org/0000-0002-3228-770X

### Ethics

This study was conducted in strict accordance with local animal ethics procedures as approved by the University of Sydney Ethics Committee (Project number: 019/1642).

### Decision letter and Author response

Decision letter https://doi.org/10.7554/eLife.76262.sa1
Author response https://doi.org/10.7554/eLife.76262.sa2

## Additional files

### Supplementary files

• Transparent reporting form

### Data availability

All data generated or analysed during this study are included in the manuscript and supporting files.

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
