## [Editor Report]

The authors study insulin secretion in acutely prepared pancreatic slices and find that it is remarkably different from what is observed in isolated islets. In particular, the work shows that polar differentiation and higher stimulus-secretion coupling is caused by integrin-mediated and local establishment of effective release sites where the beta cells contact the capillaries, involving concentration of active zone proteins and clustering of calcium channels. The findings are important and should be of significant interest to the part of readership with an interest in the regulation of exocytosis in general and insulin secretion in particular.

---

## [Decision Letter]

**Decision letter after peer review:**

Thank you for submitting your article "Local activation of focal adhesion kinase orchestrates the positioning of presynaptic scaffold proteins and ca^2+^ channel function to control glucose dependent insulin secretion." for consideration by *eLife*. Your article has been reviewed by 3 peer reviewers, including Reinhard Jahn as Reviewing Editor and Reviewer #1, and the evaluation has been overseen by Richard Aldrich as the Senior Editor.

As you can see from the enclosed comments, the reviewers differed in their opinion about the manuscript. They have discussed the relevant issues and have agreed on revisions that they consider as essential, and the Reviewing Editor has drafted this to help you prepare a revised submission. The reviewers realize that these revisions may require substantial additional work, but considering that all methods are established in your laboratory they are considered as doable during revision. Please note that also the individual comments of the reviewers should be addressed in your response

Essential revisions:

1. The authors should perform comparative experiments on freshly isolated islets in order to exclude that the reduced response of the islets is not due to the extended culture of the isolated islets.

2. The authors should check whether islets are less sensitive than slices to KATP-channel blockers as reduced responsiveness may be causally involved in the lower glucose sensitivity.

3. It would strengthen the work considerably if experiments using the approach shown in Figure 2 (measurements of exocytosis using sulforhodamine B) were included in which the response of isolated islets and slices is compared in the presence or absence of the FAK.

4. The authors should be more balanced in their discussion of their main findings, particularly with respect to their own previous publications in which the authors showed polarized insulin secretion in cultured islets being associated with local clusters of active zone proteins. Also, the summary should be revised to highlight the conceptual advance with respect to the role of the extracellular matrix in organizing exocytotic sites.

*Reviewer #1 (Recommendations for the authors):*

The labelling of the figures is not optimal, partially sloppy, and lacking important details. For instance, in all figures comparing isolated islets with slices it should be clearly indicated which panels refer to which system instead of forcing the reader to decipher the subpanels from the partially confusing legends (particularly important in Figures 1, 3 and 4).

Additional points:

Figure 1 Labelling of Figure 1C and D is confusing (top: black box slice and grey box isolated islet probably refers to C only?). Also, details are missing as to how exactly the fluorescence signals were normalized and quantified. The heatmaps are completely unintelligible: What is meant with "total surface area"? Or with the term "focussed enrichment" (at the capillary interface?)? Heatmaps: How were these maps obtained (from a 3D projection or from single confocal images?), and was there any averaging of images? Are the black bars supposed to be scale bars (length?) of 2D-images?

Figure 2 A: Is this staining for lamin by which the capillary interfaces were identified? This is neither mentioned in the text nor in the figure legend.

*Reviewer #2 (Recommendations for the authors):*

The hypersensitivity to glucose – not seen by other groups and which is NOT a sign of a more physiological preparation (on the contrary!) – has to be clarified. Independent functional assays should be performed as to the status of the preparations (e.g., lactate levels, other peptide hormone secretions, ATP levels, apoptosis assays etc) to find valid explanations.

Statistics are erroneous or insufficient in several instances. Student's t-test should not be used in Figure 2F, 6E, 8D (multiple comparisons conditions, one should not just pick that one wishes to compare and disregard the others). It is often not clear what "n" means, different slices from same animals or from different animals? Often the numbers of replica are limit especially in view of the well-known islet β-cell heterogeneity. It would also be interesting to provide some information on islet sizes, whether images were taken (at centre or periphery, where heterologous cell contacts differ and β-cells are known to have distinct behaviours).

Kinetics of stimuli arrival has only been determined for islets (Suppl Figure 4), but not for the slices, which is far more crucial.

*Reviewer #3 (Recommendations for the authors):*

I liked this paper. The difference between insulin secretion/exocytosis in isolated islets and the slices is quite remarkable.

1) Line 162 – How quickly do the changes between isolated islets and the slices develop? The methods suggest that the isolated islets were maintained in tissue culture (line 597--) but I could no information on the duration of the culture. This should be rectified. It seems unlikely that the redistribution occurs immediately upon isolation and whilst most laboratories use islets after tissue culture, some laboratories use acutely isolated islets. Such information would guide experimental work and may explain some contradictory data in the literature!

2) Line 344. How selective is Y15? If the authors are correct then it would seem that Y15 should NOT be effective in isolated cultured islets – Is this the case? This simple experiment would go a long way to establishing the specificity of the inhibitor!

3) Line 435: 'an action of F-actin'. Actin is mentioned here but not commented on subsequently. There is an interesting and potentially highly relevant effect of F-actin on KATP channel activity (PMID: 19213846). One possible explanation for the left-shift in the glucose dependence is that KATP channel activity is reduced. This possibility should be discussed. Indeed, it would be very interesting to compare the effects of cytochalasin D on insulin secretion in isolated islets and the slices (expected to have a particularly strong effect at low glucose in slices and possibly even restoring the amplifying effect of glucose in this preparation).

4) Are there any data suggesting that the observations the authors describe here can be extended to the human pancreas? Considering the work of Solimena and colleagues might be illuminating in this context (PMID: 32268101).

5) Broader significance. The first part (line 547-) is relevant and interesting. The part discussing tyupe-1 diabetes (line 561 onwards) a bit construed. Cut?

[Editors’ note: further revisions were suggested prior to acceptance, as described below.]

Thank you for resubmitting your work entitled **"**Local activation of focal adhesion kinase orchestrates the positioning of presynaptic scaffold proteins and ca^2+^ signalling to control glucose dependent insulin secretion**"** for further consideration by *eLife*. Your revised article has been seen again by the referees.

The manuscript has been improved but two of the reviewers have made further suggestions that we would like you to consider when preparing the final version for publication. Please note that we do not consider it as necessary to carry out additional experiments for the manuscript to become acceptable for publication.

*Reviewer #2 (Recommendations for the authors):*

The authors have provided additional experiments which were suggested, corrected statistical errors, and down-tuned some parts. The criticisms raised have thus been fully addressed.

It is a bit regrettable that the discussion on glucose sensitivity is still, sorry to say so, quite lack lustre and does not answer the main issue – is that physiological or an artifact (albeit in the noble sense of the term), do islet β-cells have spontaneous activity? For human studies, there are some old clinical data and recent CGM data suggesting this (activity at normoglycemia). The (correct) citation of former work from JC Henquin's laboratory does not answer that point.

It is also noteworthy that data on islet β-cell activity have been published with a maximal effect at glucose of 9 mM (as here in slices), thus the maximal and EC50 observed in slices are quite comparable to a number of islet studies.

Some improvement here may be of interest for the authors as the manuscript may become a quite often cited article.

*Reviewer #3 (Recommendations for the authors):*

The revised version of the manuscript is improved over the original but some weaknesses remain.

1) I accept that it may seem beyond the scope of the paper to explore the interaction between KATP channels and the actin network (but I hope the authors intend to carry out these measurements and that it is not just lip service to placate the reviewer!). Ironically the manuscript already contains one piece of data that strongly suggests that KATP channel activity is reduced in slices: viz. on ll. 274-, the authors mention the different effects of glibenclamide in slices and intact islets and that the KATP channel inhibitor only stimulates insulin secretion at 8 mM glucose in isolated islets (whereas stimulation is observed at 2.8 mM glucose in both preparations). I posit that this is because the KATP channels are already fully inhibited at 8 mM glucose in slices but not in isolated islets! These data are highly relevant to the manuscript's narrative and should not be hidden in the supplementary material.

2) I still miss information in Figure 3D,H about when glucose was added.

3) Given the discussion above on the role of the KATP channels, it would also be valuable to compare the effects of 8 mM glucose on cytoplasmic ca^2+^ (rather than the saturating concentration of 16.7 mM that is now used in Figure 4A-D.

4) It can be discussed how meaningful it is to compare wave velocity at 2.8 m glucose in isolated islets given that there were only 3 active cells,

5) It seems that culturing β cells on laminin does not fully rescue the 'phenotype' as no calcium oscillations were observed at low glucose (assumed to be 2.8 mM; not indicated) (Figures 7I-J and 8A-B). Comment!

6) I was pleased to see that the authors thought it was a good idea to test Y15 in isolated islets but unfortunately I think 3 replicates is not sufficient to allow any firm conclusions – there is a tendency that Y15 inhibit insulin secretion! Also, I notice that the fractional insulin release (% of content) is much higher in this experiment than in Figure 1E. This experiment should be repeated to increase the robustness of the statistical analysis (Sorry!)

7) Line 429: It is intriguing that the effects of Y15 on insulin secretion are 'reversible'. Although the figure legends remain very verbose and read like results, they lack the essential information how quickly the effects of Y15 are reversed? Does this correlate with redistribution of ELKS and liprin?

---

## [Author Response]

Essential revisions:1. The authors should perform comparative experiments on freshly isolated islets in order to exclude that the reduced response of the islets is not due to the extended culture of the isolated islets.

In response we have directly compared freshly isolated islets with islets cultured overnight. In isolated islets we observed no statistical differences in insulin secretion measured immediately compared to overnight culture either at 2.8 mM glucose (Student t test, p=0.57, n=3 mice) or at 16.7 mM glucose (Student t test, p=0.46, n=3 mice). We note that overnight culture of islets is normal practice and would not be considered in the field as “extended” culture. As explained below, in all our experiments, both isolated islets and slices, we use the same overnight culture protocol. It would be difficult to time-match measurements on fresh slices with isolated islets since preparing each individual slice is time consuming. Placing both preparations in culture under exactly the same conditions overnight, we believe, is therefore the best approach.

2. The authors should check whether islets are less sensitive than slices to KATP-channel blockers as reduced responsiveness may be causally involved in the lower glucose sensitivity.

In response we have used the KATP channel blocker glibenclamide on isolated islets and on slices. At low glucose (2.8 mM) pretreatment with glibenclamide significantly (Student t test p<0.01) increased insulin secretion in both islets and slices (new Supplemental Figure 2C), indicating that both preparations are responsive to KATP channel blockade and, by inference, that β cells are functional in both preparations.

As an additional experiment we also pretreated with glibenclamide and then applied a suprathreshold glucose concentration of 8 mM. In the isolated islets this further, significantly (Student t test p<0.01) increased insulin secretion, where-as in slices there was no further increase. This result is consistent with our findings with diazoxide, a KATP channel opener (Figure 2), and supports the idea that glucose-dependent triggering, which is dependent on closure of the KATP channel is more robust in slices compared to islets. This data is now included in the Results section (new Supplemental Figure 2C).

3. It would strengthen the work considerably if experiments using the approach shown in Figure 2 (measurements of exocytosis using sulforhodamine B) were included in which the response of isolated islets and slices is compared in the presence or absence of the FAK.

We agree and have now conducted these experiments. As expected from our observed reduction in glucose dependent ca^2+^ responses and secretion (new Figure 6 F,G,H,I), the inhibition of FAK reduces the overall number of granule fusion events. But, consistent with a role of integrin/FAK in localising granule fusion events our analysis shows the remaining events are not targeted to the capillary interface.

4. The authors should be more balanced in their discussion of their main findings, particularly with respect to their own previous publications in which the authors showed polarized insulin secretion in cultured islets being associated with local clusters of active zone proteins. Also, the summary should be revised to highlight the conceptual advance with respect to the role of the extracellular matrix in organizing exocytotic sites.

We have rewritten the discussion to include a direct assessment of our own previous data and underscore the advance in this new paper. Live-cell imaging of slices has been challenging and the high-resolution, subcellular imaging by us (eg Low et al., 2014) and others (eg Ohara-Imaizumi et al., 2019) has therefore previously been performed using isolated islets. However, fixed slices have been a valuable tool for 3D immunostaining and are the basis for our detailed localisation of active zone proteins (Low et al., 2014) and integrins (Gan et al., 2018) at the capillary interface. Thus the direct comparison of data from both live and fixed-cell work in islets and slices, has not been possible before and is a key advance in our new paper (Figure 1 and 2).

To address the second point, we now speculate more broadly on the potential mechanisms that link integrin/FAK to active zone proteins and the local organisation of exocytosis. Our new data suggests either liprin and/or ELKS are physically coupled to integrins at the capillary interface. We now discuss precedent, in other systems, particularly cortical microtubule stabilising complexes, for mechanisms that co-locate these proteins to focal adhesions in non-secretory cells.

Reviewer #1 (Recommendations for the authors):The labelling of the figures is not optimal, partially sloppy, and lacking important details. For instance, in all figures comparing isolated islets with slices it should be clearly indicated which panels refer to which system instead of forcing the reader to decipher the subpanels from the partially confusing legends (particularly important in Figures 1, 3 and 4).

We have added detail to the figure labelling to identify the details of the conditions in the panels/subpanels.

Additional points:Figure 1 Labelling of Figure 1C and D is confusing (top: black box slice and grey box isolated islet probably refers to C only?). Also, details are missing as to how exactly the fluorescence signals were normalized and quantified. The heatmaps are completely unintelligible: What is meant with "total surface area"? Or with the term "focussed enrichment" (at the capillary interface?)? Heatmaps: How were these maps obtained (from a 3D projection or from single confocal images?), and was there any averaging of images? Are the black bars supposed to be scale bars (length?) of 2D-images?

Labelling of 1C and D changed. Details of quantification added to methods. We have previously published on the use of heat maps to give a 2D representation of the 3D surface area of an individual β cell within slices (Gan et al., 2018). For clarity, this methodology is now outlined in the figure legend and included in the methods of the revised manuscript. The black bars are scale bars of 5 µm length.

Figure 2 A: Is this staining for lamin by which the capillary interfaces were identified? This is neither mentioned in the text nor in the figure legend.

The staining is the fluorescent signal from SRB. When SRB is added to the extracellular media surrounding the slices/islets it enters and highlights the capillaries and the narrow spaces between the cells (as shown in Figure 2A and B). We have previously tested the validity of this method for identification of capillaries by counterstaining with isolectin B4 (Low et al., 2014) and now explain this in the figure legends of the revised manuscript.

Reviewer #2 (Recommendations for the authors):The hypersensitivity to glucose – not seen by other groups and which is NOT a sign of a more physiological preparation (on the contrary!) – has to be clarified. Independent functional assays should be performed as to the status of the preparations (e.g., lactate levels, other peptide hormone secretions, ATP levels, apoptosis assays etc) to find valid explanations.

We understand the referee’s reservations and indeed we were also surprised when initially presented with the first results. What we can say is:

– The data has been reproduced over the last 2 years in the lab by multiple personnel.

– It is difficult to compare our data with other work on slices. It is only our detailed doseresponse to glucose that demonstrates a shift in the sensitivity to glucose: a simple low-high glucose assay will not show any change.

– Along the same lines, increased secretory output in slices is evident in our study (Figure 2) but would not be apparent in any stand-alone study of slices. Our data directly compares slices and islets under the same conditions – this has not been done before

– Our use of diazoxide, the KATP channel opener, is a selective manipulation of β cells and shows a significant difference in responses between islets and slices. This proves that β cells are central to the different responses in slices vs islets. We have used this as evidence for the enhanced preservation of structure and function of β cells in slices. If the referee was correct then the “defective” response in the slices must be due to damage to the β cells.

– We used Hoechst and propidium iodide staining to identify compromised cells in both preparations. In isolated islets we identified 11% of a total of 329 cells as stained with propidium iodide. These cells were peripheral to the islets and likely arise due to the isolation process. In slices we identified 2% of 249 cells as positive for propidium iodide. This demonstrates that cells within the slice preparation are healthy.

Statistics are erroneous or insufficient in several instances. Student's t-test should not be used in Figure 2F, 6E, 8D (multiple comparisons conditions, one should not just pick that one wishes to compare and disregard the others). It is often not clear what "n" means, different slices from same animals or from different animals? Often the numbers of replica are limit especially in view of the well-known islet β-cell heterogeneity. It would also be interesting to provide some information on islet sizes, whether images were taken (at centre or periphery, where heterologous cell contacts differ and β-cells are known to have distinct behaviours).

We have revised the analysis to include ANOVA (and Tukey tests) where relevant and clarified the meaning of the numbers in terms of replicates, islets and animals.

Kinetics of stimuli arrival has only been determined for islets (Suppl Figure 4), but not for the slices, which is far more crucial.

We measured the kinetics in exactly the same way using a dye tracer for both islets and slices. The Suppl Figure 4 shows an exemplar trace in islets, but the method was used in all the experiments. We note, our measures of latency, using a dye tracker, are much more accurate than most other methods. We applied this method to both islets and slices and therefore we have a direct “internal” comparison. The figure legend has been modified to make this point clear.

Reviewer #3 (Recommendations for the authors):I liked this paper. The difference between insulin secretion/exocytosis in isolated islets and the slices is quite remarkable.1) Line 162 – How quickly do the changes between isolated islets and the slices develop? The methods suggest that the isolated islets were maintained in tissue culture (line 597--) but I could no information on the duration of the culture. This should be rectified. It seems unlikely that the redistribution occurs immediately upon isolation and whilst most laboratories use islets after tissue culture, some laboratories use acutely isolated islets. Such information would guide experimental work and may explain some contradictory data in the literature!

We use both slices and islets after overnight culture – this is now clarified in the text. As a point of interest we have some pilot data that show islet isolation leads to a collapse of β cell structure within hours. However, this can only be qualitatively assessed because enzyme digestion has variable effects from mouse-to-mouse, big vs small islets etc. So, although all islets are damaged and capillaries are lost after isolation, the extent of this is variable.

2) Line 344. How selective is Y15? If the authors are correct then it would seem that Y15 should NOT be effective in isolated cultured islets – Is this the case? This simple experiment would go a long way to establishing the specificity of the inhibitor!

Y15 is a selective blocker of FAK Y397 phosphorylation and, while we cannot rule out side effects, the consistency of our observed responses to Y15, and to integrin blockade and to culture of cells on BSA all point to an integrin/FAK pathway.

3) Line 435: 'an action of F-actin'. Actin is mentioned here but not commented on subsequently. There is an interesting and potentially highly relevant effect of F-actin on KATP channel activity (PMID: 19213846). One possible explanation for the left-shift in the glucose dependence is that KATP channel activity is reduced. This possibility should be discussed. Indeed, itwould be very interesting to compare the effects of cytochalasin D on insulin secretion in isolated islets and the slices (expected to have a particularly strong effect at low glucose in slices and possibly even restoring the amplifying effect of glucose in this preparation).

This is an interesting point, the KATP channel might be enriched at the capillary interface (because it is linked with SNARES – Leung et al., 2007) and this might be better preserved in the slices. However, we think that this would justify a separate study since F-actin plays multiple roles in the β cell, (for example, we have recently published on the local action of F-actin in controlling insulin granule dynamics during fusion (Wei et al., 2020)) and therefore interpretation of findings of inhibitors would be difficult.

4) Are there any data suggesting that the observations the authors describe here can be extended to the human pancreas? Considering the work of Solimena and colleagues might be illuminating in this context (PMID: 32268101).

We have now included reference to Solimena’s work also reference to our own work on human pancreas which shows human β cells possess these active zone proteins and they are enriched at the capillary interface (Cottle et al., 2021). Importantly, our paper shows that despite some gross morphological differences between mouse and human islets each individual β cell in the human islet still makes contact with an islet capillary.

5) Broader significance. The first part (line 547-) is relevant and interesting. The part discussing tyupe-1 diabetes (line 561 onwards) a bit construed. Cut?

We would like to leave the discussion of the relevance to cell-based approaches to treating type 1 diabetes. In that field there is a view that β cell function is defined by pathways intrinsic to the cell and that external factors might enhance insulin secretion but are not fundamentally important. What our work suggests is that external factors might be much more important than previously thought and we think that a short discussion of this point is relevant.

[Editors’ note: further revisions were suggested prior to acceptance, as described below.]

Reviewer #2 (Recommendations for the authors):The authors have provided additional experiments which were suggested, corrected statistical errors, and down-tuned some parts. The criticisms raised have thus been fully addressed.

We thank the referee for their considered comments.

It is a bit regrettable that the discussion on glucose sensitivity is still, sorry to say so, quite lack luster and does not answer the main issue – is that physiological or an artifact (albeit in the noble sense of the term), do islet β-cells have spontaneous activity? For human studies, there are some old clinical data and recent CGM data suggesting this (activity at normoglycemia). The (correct) citation of former work from JC Henquin's laboratory does not answer that point.

The reality is that we do not know whether the insulin secretion we observe in slices is “more” physiological. Data from humans and animals does show insulin secretion even at low glucose concentrations and we have now added discussion around this point. It is interesting that insulin secretion at low (fasting levels) of glucose is consistently observed as oscillatory since this could be driven by the spikes of ca^2+^ we observe at low glucose concentrations. However, we are not aware of any in vivo data showing insulin secretion at lower than normal endogenous glucose concentrations. To do this would be difficult and require deliberate and careful experimentation and would be complicated by the triggering of glucagon secretion.

It is also noteworthy that data on islet β-cell activity have been published with a maximal effect at glucose of 9 mM (as here in slices), thus the maximal and EC50 observed in slices are quite comparable to a number of islet studies.

We agree with this sentiment and have added a sentence in the discussion to reflect this.

Some improvement here may be of interest for the authors as the manuscript may become a quite often cited article.Reviewer #3 (Recommendations for the authors):The revised version of the manuscript is improved over the original but some weaknesses remain.1) I accept that it may seem beyond the scope of the paper to explore the interaction between KATP channels and the actin network (but I hope the authors intend to carry out these measurements and that it is not just lip service to placate the reviewer!). Ironically the manuscript already contains one piece of data that strongly suggests that KATP channel activity is reduced in slices: viz. on ll. 274-, the authors mention the different effects of glibenclamide in slices and intact islets and that the KATP channel inhibitor only stimulates insulin secretion at 8 mM glucose in isolated islets (whereas stimulation is observed at 2.8 mM glucose in both preparations). I posit that this is because the KATP channels are already fully inhibited at 8 mM glucose in slices but not in isolated islets! These data are highly relevant to the manuscript's narrative and should not be hidden in the supplementary material.

We thank the referee for their original comments which provoked us to use glibenclamide. The findings are entirely consistent with our original data using the KATP channel opener diazoxide. We agree with the referee that this is important additional information that reinforces the idea that the glucose-dependent trigger is the focus of the differences between the slice and the isolated islets. We have therefore now moved the data to a new Figure 3 C.

2) I still miss information in Figure 3D,H about when glucose was added.

We have now added the time bar (in new Figure 4) to indicate the point of addition of high glucose (as determined by the SRB fluorescence tracer).

3) Given the discussion above on the role of the KATP channels, it would also be valuable to compare the effects of 8 mM glucose on cytoplasmic ca^2+^ (rather than the saturating concentration of 16.7 mM that is now used in Figure 4A-D.

We agree with this sentiment and think that further exploration to further determine the basis of the glucose dose-dependence we reveal in slices will be interesting. However, we think this is for future studies and that the current paper already makes significant advances with a clear narrative centred on β cell organisation, novel ca^2+^ responses and new insights into the regulatory roles of the integrin/FAK pathway.

4) It can be discussed how meaningful it is to compare wave velocity at 2.8 m glucose in isolated islets given that there were only 3 active cells,

We think this is already clearly stated as a limitation in the text. Of course, it reflects the real-life issue that waves were only rarely observed in slices.

5) It seems that culturing β cells on laminin does not fully rescue the 'phenotype' as no calcium oscillations were observed at low glucose (assumed to be 2.8 mM; not indicated) (Figures 7I-J and 8A-B). Comment!

We agree that the in vitro culture on ECM proteins does not fully recapitulate our findings in slices. We have added a statement to this effect line 411.

6) I was pleased to see that the authors thought it was a good idea to test Y15 in isolated islets but unfortunately I think 3 replicates is not sufficient to allow any firm conclusions – there is a tendency that Y15 inhibit insulin secretion! Also, I notice that the fractional insulin release (% of content) is much higher in this experiment than in Figure 1E. This experiment should be repeated to increase the robustness of the statistical analysis (Sorry!)

We did think it was a good idea and thank the referees for this in the original comments. We also accept that the data is variable. However, similar variability is seen in the slice data (see Figure 7E) with the difference, compared to isolated islets, being the larger effect of Y15 on the mean response. So, with similar n numbers and similar variability, we see a significant effect of Y15 on insulin secretion in slices and not in isolated islets.

To directly answer the referee’s question, we have performed a Power calculation based on the data for the current 3 replicates in isolated islets. If there was a real difference in the means, and Y15 was inhibiting the response in isolated islets, then the calculations are that to show significance at an α of 0.05 would require 141 replicates which is beyond the scope of our study.

7) Line 429: It is intriguing that the effects of Y15 on insulin secretion are 'reversible'. Although the figure legends remain very verbose and read like results, they lack the essential information how quickly the effects of Y15 are reversed? Does this correlate with redistribution of ELKS and liprin?

They were reversed after a 6 hour wash in control solution. We did not test for the repositioning of ELKS and liprin but agree this would be an interesting experiment.